

# Modeling the impact of solar "brightening" on summer surface ozone over Europe between 1990 and 2010

Emmanouil Oikonomakis[1], Sebnem Aksoyoglu[1], Martin Wild[2], Giancarlo Ciarelli[3], Urs Baltensperger[1] and André S. H. Prévôt[1]

[1]Laboratory of Atmospheric Chemistry, Paul Scherrer Institute, Villigen, 5232, Switzerland
[2]Institute for Atmospheric and Climate Science, Swiss Federal Institute of Technology (ETH), Zurich, Switzerland
[3]Laboratoire Inter-Universitaire des Systèmes Atmosphériques (LISA), UMR CNRS 7583, Université Paris Est Créteil et Université Paris Diderot, Institut Pierre Simon Laplace, Créteil, France

*Correspondence to*: Sebnem Aksoyoglu (sebnem.aksoyoglu@psi.ch)

## Abstract

Surface solar radiation (SSR) observations have indicated an increasing trend in Europe since the mid-1980s, referred to as solar "brightening". In this study, we used the regional air quality model, CAMx (Comprehensive Air Quality Model with extensions) to simulate and quantify, with various sensitivity runs (where the year 2010 served as the base case), the effects of increased radiation on photolysis rates (PHOT1, PHOT2 and PHOT3 scenarios) and biogenic volatile organic compounds (BVOCs) emissions (BIO scenario), and their consequent impacts on summer surface ozone concentrations over Europe between 1990 and 2010. The PHOT1 and PHOT2 scenarios examined the effect of doubling and tripling the anthropogenic $PM_{2.5}$ concentrations, respectively, while the PHOT3 investigated the impact of an increase in just the sulfate concentrations by a factor of 3.4 (as in 1990), applied only to the calculation of photolysis rates. In the BIO scenario, we reduced the 2010 SSR by 3% (keeping plant cover and temperature the same), re-calculated the biogenic emissions and repeated the base case simulations with the new biogenic emissions. The impact on photolysis rates for all three scenarios was an increase (in 2010 compared to 1990) of 3-6% which resulted in daytime (10:00–18:00 Local Mean Time (LMT)) mean surface ozone differences of 0.2–0.7 ppb (0.5–1.5%), with the largest hourly difference rising as high as 4–8 ppb (10–16%). The effect of changes in BVOCs emissions on daytime mean surface ozone was much smaller (up to 0.08 ppb, ~ 0.2%), as isoprene and terpene (monoterpene and sesquiterpene) emissions increased only by 2.5–3% and 0.7%, respectively. Overall, the impact of the SSR changes on surface ozone was greater via the effects on photolysis rates compared to the effects on BVOCs emissions, and the sensitivity test of their combined impact (PHOT3+BIO = COMBO scenario) showed nearly additive effects. In addition, all the sensitivity runs were repeated on a second base case with increased $NO_x$ emissions to account for any potential underestimation of modeled ozone production; the results did not change significantly in magnitude, but the spatial coverage of the effects was profoundly extended. Finally, the role of the solar "brightening" in the European summer surface ozone trends was suggested to be more important when comparing to the order of magnitude of the ozone trends





instead of the total ozone concentrations, indicating a potential partial damping of the effects of ozone precursor emissions reduction.

**Introduction**

Solar radiation plays a key role in the atmospheric chemistry by photo-dissociation of gas-molecules. Photolysis reactions, which are mainly driven by the ultraviolet part of the spectrum (100–400 nm), have a significant impact on the formation of tropospheric air pollutants like ozone (Madronich and Flocke, 1999; Seinfeld and Pandis, 2016). The photolysis of ozone leads to its self-destruction (R1) and in the presence of water vapor it becomes the main source of hydroxyl radicals (OH) in the troposphere (R2), while the photolysis of $NO_2$ will lead to ozone production via reaction R3 and R4 (Madronich and Flocke, 1999; Monks, 2005):

$$O_3 + hv \ (\lambda < 320 \ nm) \xrightarrow{J_{O_3 \rightarrow O^1D}} O(^1D) + O_2 \tag{R1}$$

$$O(^1D) + H_2O \rightarrow OH + OH \tag{R2}$$

$$NO_2 + hv \ (\lambda < 420 \ nm) \xrightarrow{J_{NO_2}} NO + O(^3P) \tag{R3}$$

$$O(^3P) + O_2 + M \rightarrow O_3 + M \tag{R4}$$

The photolysis rate coefficient ($J$) of a gas is wavelength ($\lambda$) dependent and is described by the following equation (Madronich and Flocke, 1999) :

$$J = \int F(\lambda) \cdot \varphi(\lambda, T) \cdot \sigma(\lambda, T) \ d\lambda \tag{1}$$

where $F$ is the solar actinic flux (photons $cm^{-2} \ s^{-1} \ nm^{-1}$) which represents the solar radiation that is incident to a volume element, and $\varphi$ and $\sigma$ are the quantum yield and absorption cross section ($cm^2$), respectively, of the gas. $\varphi$ and $\sigma$ depend on the gaseous species and on the air temperature $T$ (K), while $F$ depends on the position of the sun and the transmissivity of the atmosphere which is mainly influenced by the presence of clouds, aerosols and radiatively active gases (e.g. water vapor). Since the atmosphere can be considered as an optical medium, the total light extinction is governed by the optical depth of the clouds (COD) which mainly scatter light and of the aerosols (AOD) which either scatter or absorb light (aerosol direct radiative effects, DRE) depending on their optical properties (Yu et al., 2006; Seinfeld and Pandis, 2016), as well as by the absorption of gases. In addition, aerosols have an indirect influence on the atmospheric transmissivity (aerosol indirect effects, AIE) as they play a role in the formation of clouds by serving as cloud condensation nuclei (CCN) and they can also alter the optical properties and lifetime of clouds (Lohmann and Feichter, 2005; Seinfeld and Pandis, 2016). Aerosols are either directly emitted (primary aerosols) by anthropogenic (e.g. industries, heating processes, vehicles, ships, biomass burning) and natural sources (e.g. volcanos, oceans, deserts), or they are formed through chemical reactions (secondary



aerosols) of precursor gases, i.e. $SO_2$, $NH_3$, nitrogen oxides ($NO_x=NO_2+NO$), volatile organic compounds (VOC) (Fuzzi et al., 2015; Seinfeld and Pandis, 2016). Hence, the human activities can affect the incoming solar radiation by influencing the aerosol loading and radiatively active gas concentrations in the atmosphere.

The multi-decadal changes in aerosol concentrations in the 20[th] century are considered to be responsible for the changes in
surface solar radiation (SSR) in several areas in western Europe and North America: There was a decrease in the SSR between 1950s and mid-1980s (referred to as solar "dimming") due to increased industrial and urban production of aerosols, followed by an increase in the SSR since the mid-1980s (referred to as solar "brightening") when air quality regulations were imposed (Stanhill and Cohen, 2001; Wild et al., 2005; Streets et al., 2006; Ohmura, 2009; Wild, 2009, 2012; Allen et al., 2013; Imamovic et al., 2016). Moreover, extraterrestrial changes or changes in radiatively active gases were ruled out as
potential drivers of the solar "dimming" and "brightening" (Kvalevåg and Myhre, 2007; Wild, 2009). On the other hand, there are studies arguing that these changes in SSR, especially in pristine or remote areas, were mainly driven by natural changes in cloud cover and/or cloud properties (Dutton et al., 2006; Long et al., 2009; Augustine and Dutton, 2013; Stanhill et al., 2014). However for Europe, several studies have reported either no statistically significant trends in cloud cover since 1990 or no strong evidence that changes in cloud cover were mainly responsible for the observed SSR trends (Norris and
Wild, 2007; Sanchez-Lorenzo et al., 2009, 2012, 2017a; Vetter and Wechsung, 2015). Furthermore, other studies that focused on Europe pointed to aerosols, and especially the direct effect, as the main driver for the brightening since the mid-1980s (Ruckstuhl et al., 2008, 2010; Ruckstuhl and Norris, 2009; Folini and Wild, 2011; Sanchez-Lorenzo and Wild, 2012; Wang et al., 2012b; Cherian et al., 2014; Nabat et al., 2014; Turnock et al., 2015; Manara et al., 2016). The relative contribution of clouds and aerosols to the SSR trends might also have a seasonal and spatial dependence, which could be
related to changes in large scale atmospheric circulation patterns like the North Atlantic Oscillation (Stjern et al., 2009; Chiacchio and Wild, 2010; Chiacchio et al., 2011; Parding et al., 2016), or can also depend on the method of study, e.g. surface measurements, satellite observations, SSR proxies like sunshine duration (Sanchez-Lorenzo et al., 2008, 2017b). In addition, it is not clear yet if and to what extent aerosol-cloud interactions influenced the SSR trends in Europe since the mid-1980s (Wild, 2009; Ruckstuhl et al., 2010; Boers et al., 2017).

Tropospheric ozone in Europe has either not decreased as much as expected or even increased in spite of large reductions of precursor emissions since the 1990s (Wilson et al., 2012; Aksoyoglu et al., 2014; Colette et al., 2016). In addition to precursor emissions, European ozone concentrations might also be affected by the hemispheric baseline ozone and changes in photochemical activity (Ordóñez et al., 2005; Andreani-Aksoyoğlu et al., 2008). The radiative impact of aerosols on photochemistry and tropospheric ozone over Europe has been examined in several studies (Real and Sartelet, 2011; Forkel et
al., 2012, 2015; Kushta et al., 2014; Makar et al., 2015a; San José et al., 2015; Xing et al., 2015a; Mailler et al., 2016). Real and Sartelet (2011) used an offline model (where meteorology and chemistry are decoupled) and performed simulations with and without DRE. They reported that the photolysis rates at the ground level were reduced in the summer, due to the DRE, by 10-14% which led to an average surface ozone reduction of 3% and up to 8% in more polluted areas. A different approach was followed by Xing et al. (2015a,b) to investigate and quantify the impact of multi-decadal DRE changes on



surface ozone between 1990 and 2010 over the Northern Hemisphere, by using an online-coupled model. For Europe, they reported a total average increase of 0.3% and up to 3% for more polluted days throughout 21 years when they included the DRE (compared to the no-feedback case). In other words, they suggested that higher AOD (and thus larger DRE) led to higher ozone concentrations due to an increase of atmospheric vertical stability (lower Planetary Boundary Layer (PBL) height) as a result of the DRE surface cooling and above-PBL warming, which resulted in an increase of ozone formation by the accumulation of pollutants close to the surface. This feedback overcompensated for the decreased photolysis rates (due to solar radiation reduction by DRE), although increased photolysis rates do not always lead to higher ozone production (as discussed above), but they can also lead to higher ozone destruction in $NO_x$-limited environments (Bian et al., 2003).

On the other hand, other modeling studies for different summer periods and regions (US and Europe) showed that the DRE on ozone vary spatially, leading to either ozone enhancement or reduction depending on the local meteorological and chemical conditions (Forkel et al., 2012, 2015; Hogrefe et al., 2015; Kong et al., 2015; Makar et al., 2015a; Wang et al., 2015). Moreover, they showed that the impact on ozone (both enhancement and reduction) was even stronger when the AIE were also taken into account. In addition, Forkel et al. (2012) suggested that the spatial patterns of changes in meteorological features due to the aerosol effects should not be taken as a general feature, because they will depend on the prevailing meteorological conditions. Makar et al. (2015a,b) further pointed out that the modeling results of AIE on weather (and consequently on chemistry) will vary based on the model parameterization when comparing the no-feedback case (some models use a "no aerosol" atmosphere while some other use different simple parameterizations for aerosol radiative properties and CCN formation) with the one including the AIE. Overall, the research about the aerosol radiative effects (especially the indirect effects) and their implementation in the online-coupled models to consistently simulate their feedbacks on meteorology and chemistry is still going on, along with the efforts to overcome the problems of high computational demand (Zhang, 2008; Baklanov et al., 2014).

The focus of this study was to investigate the impact of changes in solar radiation in Europe between 1990 and 2010 on summer surface ozone with the following main differences from pre-existing studies: First, we used an offline model; thus excluding AIE, following suggestions from several studies that the DRE was the main driver for the brightening in Europe during the period 1990–2010. In this way, we also excluded the meteorological feedbacks on chemistry due to DRE, emphasizing the more direct and less uncertain impact of DRE on chemistry via the photolysis rates, compared to the more uncertain meteorology-chemistry interactions in the online-coupled models as discussed above. Second, we designed specific sensitivity tests to simulate, as consistently as possible, the observed changes in AOD and SSR in Europe between 1990 and 2010, which is different from the general "switch on/off" DRE approach. Third, we modeled and compared only the initial (1990) and final year (2010) of the studied period using same model input (i.e., the one of 2010; thus the actual year 1990 was not simulated, but rather the AOD and SSR conditions representative of the 1990's; see Sect. 2.3) to isolate the DRE on ozone from other factors. Furthermore, this approach is unaffected by any potential masking of DRE on ozone from inter-annual variability of key ozone influencing factors (such as meteorology, emissions and boundary conditions), compared to multi-year (with "switch on/off" DRE) simulation studies. Fourth, we included and investigated for the first



time (to the best of our knowledge) the impact on biogenic emissions and their effects on ozone. The methods and design of the aforementioned sensitivity tests are described in Sect. 2, accompanied by a particulate matter (PM) trend analysis and discussion (where the model runs were based on) in Sect. 3. The model results are presented and discussed in Sect. 4. Finally, the conclusions are summarized in Sect. 5.

## 2. Methodology

### 2.1 Model Setup

We used the offline (i.e. the meteorology is prescribed) regional air quality model, CAMx version 6.30 (comprehensive air quality model with extensions, http://www.camx.com). We modeled the summer season (June, July, August, JJA) in 2010 plus the last two weeks of May which were used as spin-up time. The model domain had a horizontal resolution of 0.250° x

0.125° and covered the whole Europe from 15°W to 35°E and 35°N to 70°N. The vertical extension was up to 460 hPa using 14 sigma layers. The thickness of the first layer was ~20 m but its modeled values correspond to ~10 m, as the concentrations are calculated at the mid-point of each layer. We used the CB6r2 (Carbon Bond mechanism, version 6, revision 2: Hildebrandt Ruiz and Yarwood, 2013) gas phase mechanism and we simulated the PM concentrations using a bimodal (fine/coarse) size distribution. For the inorganic thermodynamics and gas–aerosol partitioning calculations the

ISORROPIA scheme (Nenes et al., 1998, 1999) was used, while for the calculations of the organic aerosol concentrations we used the SOAP model (Strader et al., 1999). The dry deposition was calculated according to the scheme of Zhang et al. (2003). The MOZART (Model of Ozone and Related Chemical Tracers) global model data for 2010 (Horowitz et al., 2003) served as initial and boundary conditions for the chemical species. The MOZART data had a time resolution of 6 hours and were interpolated to the size and resolution of our grid using the CAMx pre-processor MOZART2CAMx (RAMBOLL-

ENVIRON, 2016). The photochemical calculations in CAMx consist of two steps. Initially, calculations of clear-sky photolysis rates are performed externally using the Tropospheric Ultraviolet and Visible (TUV) radiation model (NCAR, 2011), where a climatological aerosol profile determined by Elterman (1968) is used. Then, these rates are internally adjusted in CAMx every hour for clouds and aerosols (simulated by CAMx) as well as for pressure and temperature (Emery et al., 2010; RAMBOLL-ENVIRON, 2016). This internal adjustment is carried out only for a single representative

wavelength (350 nm), as tests against the full-science TUV indicated a difference smaller than 1% in the ratio of cloud- to clear-sky solar actinic flux for a variety of cloudy conditions (Emery et al., 2010). The COD is calculated for each model grid cell based on the approach of Genio et al. (1996) and Voulgarakis et al. (2009), while the dry extinction efficiency of the aerosol species, which is needed for the calculation of the AOD, as well as the single-scattering albedo (SSA) were provided by Takemura et al. (2002) for the wavelength of 350 nm (Table S1). These values of aerosol optical properties were provided

for sulfate, organics, soot, total dust and sea salt, and the sulfate values were extended to nitrate and ammonium (RAMBOLL-ENVIRON, 2016). The asymmetry factor for aerosols was set to have a default value of 0.61 regardless of their composition. For clouds the default values of the asymmetry factor and SSA were 0.85 and 0.99, respectively. In





addition, the 8-stream discrete ordinates scheme was used for the radiative transfer calculations compared to the more common (and computationally faster) 2-stream delta-Eddington approximation scheme, as the calculations' accuracy increases with the number of streams (Stamnes et al., 1988; Toon et al., 1989). The choice of 8 streams has been suggested to offer high accuracy (1% or better compared to 32 streams) without having a significantly higher computation cost
(Petropavlovskikh, 1995). TOMS (Total Ozone Mapping Spectrometer) data, which were provided by NASA (National Aeronautics and Space Administration; ftp://toms.gsfc.nasa.gov/pub/omi/data/), were used as input for total ozone column in both TUV and CAMx. Finally, the radiative transfer algorithms of both TUV and CAMx were modified to extract the modeled AOD and SSR data.

The required meteorological input for CAMx was generated by the WRF-ARW model (Weather Research and Forecasting
Model, version 3.7.1; Skamarock et al., 2008). Re-analysis global data, with time resolution of 6 hours and horizontal resolution of 0.72° x 0.72°, were provided by ECMWF (European Centre for Medium-Range Weather Forecasts) and served as initial and boundary conditions for WRF. Both CAMx and WRF had the same model domain and horizontal resolution. However, for the WRF runs 31 vertical layers up to 100 hPa were used instead of 14, which was the case for the CAMx runs for computational efficiency. More details about the WRF parameterization are provided in Oikonomakis et al. (2017).

For the anthropogenic emissions input we used the TNO-MACC-III emission inventory for 2010. This inventory was provided by the Netherlands Organization for Applied Scientific Research (TNO) and is an extension of the TNO-MACC-II emission inventory (Kuenen et al., 2014). More details about the TNO-MACC-III emission inventory are given in Kuik et al. (2016). The TNO European emission domain is the same as our domain but with a finer horizontal resolution (0.125° x 0.0625°). The mineral dust, sea salt and wild fire emissions are not included in the inventory. However, in the model's initial
and boundary conditions the concentrations of mineral dust and sea salt are included. For the calculation of the biogenic emissions (isoprene, monoterpenes, sesquiterpenes and soil NO) we followed the methods described by Andreani-Aksoyoglu and Keller (1995) using temperature and SSR data from the WRF output as well as land use data from the GlobCover 2005-06 land use inventory (http://due.esrin.esa.int/page_globcover.php) and the United States Geological Survey (USGS). All emissions were injected in the first model layer and were treated as area emissions. A detailed discussion and values of the
emissions used in this study are given in Oikonomakis et al. (2017).

**2.2 Observations**

The European Air Quality Database v7 (AirBase; Mol and de Leeuw, 2005) provided observational surface data for the air pollutant concentrations (http://acm.eionet.europa.eu/databases/) with an hourly time resolution, which were used for chemical model evaluation. The details of observational data treatment and the statistical methods were described in the
model evaluation part of Oikonomakis et al. (2017). Furthermore, $PM_{10}$ (particles with an aerodynamic diameter, $d < 10$ μm) and $PM_{2.5}$ ($d < 2.5$ μm) data from the AirBase database as well as from the Swiss National Air Pollution Monitoring Network (NABEL; Empa, 2010) were used for trend analysis. Switzerland and the Netherlands were chosen for the PM trend analysis as they have $PM_{10}$ data going back to 1990 and 1992, respectively. For Switzerland, the $PM_{10}$ data until 1997





are actually corrected total suspended particles (TSP) data (Empa, 2010), but they are suitable for PM₁₀ trend analysis (Barmpadimos et al., 2011). Hourly, high quality SSR data from the Baseline Surface Radiation Network (BSRN; König-Langlo et al., 2013) for 7 stations were used for model evaluation. An overview of the 7 BSRN stations is given in Table S2. Finally, AOD data were retrieved by the Aerosol Robotic Network (AERONET), which is a network of ground-based sun

photometer measurements of aerosol optical properties (Holben et al., 1998; O'Neill et al., 2003). We used level 2.0 (quality assured) data for the 340 nm wavelength band to compare with the respective modeled AOD values. The calibration error of the AOD measurements is of the order of 0.015 (Holben et al., 1998; Eck et al., 1999). Since the temporal resolution of the AOD measurements is not constant (e.g. at specific hours), the calculated daily mean does not correspond to a 24 h time interval, but to intraday time intervals with available measurements. The daily average of the modeled AOD was calculated

using only the times of available AOD measurements for each site, for a more consistent comparison between model and observations.

## 2.3 Model runs

The description of 12 model runs is shown in Table 1. We used two base case scenarios: one with the default parameterization (BASE) and a second one with increased $NO_x$ emissions (BASE_$NO_x$) which produced higher ozone

concentrations, in order to incorporate any potential underestimation of the DRE effects on ozone due to underestimated modeled ozone production as suggested by Oikonomakis et al. (2017). All sensitivity tests were performed using both base case scenarios (see Table 1).

The impact of solar radiation changes due to the DRE on ozone chemistry was investigated via two pathways: i) via impact on photolysis rates, and ii) via impact on biogenic volatile organic compounds (BVOCs) emissions. In order to quantify

these impacts we first simulated the summer of 2010, then applied sensitivity tests that would represent the radiation conditions in the summer of 1990 (i.e. different solar radiation due to DRE) and finally compared the two cases. In other words, we used the same meteorology and emissions for both cases and we designed special sensitivity tests to isolate and quantify the effect of changes in the DRE between those years on ozone concentrations.

### 2.3.1 Impact via photolysis rates

In order to quantify only the DRE, we isolated them from other aerosol effects such as via the gas-aerosol chemical interactions. For this reason, we modified the radiative transfer algorithm in CAMx (i.e. the in-line version of TUV) by applying an adjustment factor ($p_f$) in the AOD calculation to represent the aerosol concentrations in 1990, but without changing the concentrations themselves and thus excluding changes due to the chemistry. So, the adjusted AOD for $N$ vertical layers and $M$ aerosol species was calculated as shown below:



$$AOD = \sum_{j=1}^{N} \Delta z_j \cdot \sum_{i=1}^{M} \mu_{ext_i} \cdot f(RH_j) \cdot C_{ij} \cdot p_{f_i} \qquad (2)$$

where $\mu_{ext}$ is the aerosol dry extinction efficiency (see Table S1), $f(RH)$ is the relative humidity ($RH$) adjustment factor (FLAG, 2000), $C$ is the aerosol species concentration, and $\Delta z$ is the layer's thickness. The value of $p_f$ for sulfate ($SO_4^{2-}$), ammonium ($NH_4^+$), nitrate ($NO_3^-$), primary organic aerosol (POA), anthropogenic secondary organic aerosol (ASOA), elemental carbon (EC) and fine other primary aerosol (FPRM) varies with the sensitivity test, while there was no adjustment

(i.e. $p_f = 1$) for biogenic secondary aerosol (BSOA), sodium chloride (NaCl), fine (FCRS) and coarse (CCRS) crustal aerosols, and coarse other primary aerosol (CPRM). We have excluded the natural aerosols (biogenic SOA, sea salt and dust (FCRS+CCRS)) from the AOD adjustment since the anthropogenic aerosol concentrations reductions were suggested as a likely explanation for the brightening (see Sect. 1); moreover no significant change in their contribution to the AOD trends was reported (Streets et al., 2009). Although large natural aerosol contributors like volcanic eruptions (e.g. El Chichón in

1986 and Pinatubo in 1991) can introduce large spikes in the SSR time series, they do not alter the longer-term trends (Wild, 2009). In addition, we have also excluded the coarse fraction of the anthropogenic aerosols from the AOD adjustment, assuming that the fine fraction dominated the decreasing trend of the total aerosol mass (discussed in detail in Sect. 3; Barmpadimos et al., 2012; Tørseth et al., 2012). The $p_f$ values 2 and 3 (corresponding to ~ 50% and 65% reductions in PM$_{2.5}$ concentrations, respectively, in 2010 compared to 1990) for the first two sensitivity tests (PHOT1 and PHOT2, respectively,

in Table 1), were inferred by a PM trend analysis based on observations (discussed in detail in Sect. 3) and they represent an estimated range of reductions in PM$_{2.5}$ concentrations between 1990 and 2010 in Europe, i.e. PM$_{2.5}$_1990 = PM$_{2.5}$_2010 $\cdot p_f$. The assumption for PHOT1 and PHOT2 scenarios is that the estimated observed changes in PM$_{2.5}$ are the same for all species, which does not necessarily correspond to reality as some species decreased more ($SO_4^{2-}$) than others ($NH_4^+$), while for some others (EC) trends are not known as there were no measurements during the 1990s in Europe (Tørseth et al., 2012).

However, sulfate was and still is one of the single most important components that contribute to the total aerosol mass concentration in Europe (Putaud et al., 2010; Tørseth et al., 2012). Moreover, the sulfate measurements started in 1972, so its trends and changes (between -60 and -80%) are well known for our period of study (Tørseth et al., 2012; Banzhaf et al., 2015; Xing et al., 2015c; Colette et al., 2016), which are within the same range as the changes considered in PHOT1 and PHOT2 scenarios. Therefore, we consider the PHOT1 and PHOT2 scenarios to be good proxies for the purpose of this study,

at a regional scale. Furthermore, in order to investigate the impact of sulfate in more detail, we included another sensitivity test (PHOT3 scenario) where we adjusted only the sulfate concentrations in 2010 by a factor of 3.4, which represents approximately a -70% total change in sulfate concentrations between 1990 and 2010 based on the aforementioned studies. Another aspect to be considered was the anthropogenic aerosols originating directly or indirectly from ship emissions. Since marine emissions were not regulated during 1990–2010 (Eyring et al., 2005; Aksoyoglu et al., 2016) we did not adjust the

AOD (i.e., $p_f = 1$) over the sea and ocean (for PHOT1, PHOT2 and PHOT3 scenarios), where the contribution of ship



emissions to the PM$_{2.5}$ concentrations is more significant (up to 50%) compared to continental Europe (up to 10%) as shown by Aksoyoglu et al. (2016) for the summer of 2006. This way, we expect that the photolysis rate sensitivity tests will represent in general more consistently the AOD conditions of 1990, even though this approach might be conservative as the European maritime AOD trends suggest a decline (significant at the 95% or 99% level) since the early/mid-1990s
(Mishchenko et al., 2007; Cermak et al., 2010; Li et al., 2014b).

### 2.3.2. Impact via BVOCs emissions

We investigated the effect of changes in the solar radiation on biogenic VOC emissions and the subsequent impact on ozone, in two steps. The first step was to generate new biogenic emissions after decreasing the solar radiation input values in the biogenic emission model by 3% (corresponding to the SSR conditions of 1990), as the observed relative change of SSR in
Europe in the summer season between 1990 and 2009 (i.e. the SSR was 3% lower in 1990 compared to 2009) according to Turnock et al. (2015). These new emissions would correspond to 1990 conditions with respect to the SSR factor; changes in other parameters due to SSR changes, like temperature and photosynthesis as well as direct to diffuse radiation ratio, were not taken into account. The second step was to re-run CAMx with these new biogenic emissions (BIO scenario) and compare with the base case (BASE scenario, Table 1). Finally we included a scenario (COMBO) with the combined effects of
biogenic emissions (BIO scenario) and photolysis rates (PHOT3 scenario; it was chosen as it was considered to be the least uncertain scenario compared to PHOT1 and PHOT2) to assess the overall impact of solar "brightening" on surface ozone.

### 3. PM trends

As discussed in Sect. 2.3, the adjustment factor ($p_f$) used in the sensitivity tests represents the total relative change in aerosol concentrations between 1990 and 2010 for the summer season. Although for the SSR such a value was available in the
literature (Turnock et al., 2015) for a similar time period (1990–2009) as in this study for the summer season, this was not the case for the total aerosol concentrations. Therefore, we performed a trend analysis to estimate the total relative change of aerosol concentrations for the time period 1990–2010. Several studies report a decreasing trend in both PM$_{10}$ and PM$_{2.5}$ concentrations in Europe since the 1990s following the reductions in the anthropogenic emissions of PM$_{10}$, PM$_{2.5}$ and gas precursors responsible for secondary aerosol formation (EEA, 2014, 2017). Barmpadimos et al. (2012) and Tørseth et al.
(2012) suggested that the decreasing trend in PM$_{10}$ concentrations was dominated by the reductions in the PM$_{2.5}$ concentrations for the periods 1998–2010 and 2000–2009 respectively, as the aerosol coarse fraction (PM$_{10}$–PM$_{2.5}$) had either a very small decrease or in some cases even a small increase. Although Wang et al. (2012a) claimed a smaller decrease in PM$_{2.5}$ than in PM$_{10}$ during 1992-2009, this could be attributed to the difference in number and type of the sites as discussed by Fuzzi et al. (2015). Hence, for our trend analysis we assumed that the aerosol coarse fraction remained constant
throughout the period 1990–2010. Therefore, we subtracted the 2010 aerosol coarse fraction from the PM$_{10}$ concentrations of



all years to infer the $PM_{2.5}$ concentrations trend and calculate their total change over the period of study. The adjustment factors ($p_f$) were then based on the total relative changes of the estimated $PM_{2.5}$ concentrations.

The linear trends were calculated with the Theil-Sen method (Sen, 1968) and their significance was evaluated with the Mann-Kendall test (Mann, 1945; Kendall, 1948). The stations selected for the trend analysis (3 for Switzerland and 3 for the
Netherlands) fulfilled the following criteria: i) they covered the whole period 1990–2010 (Switzerland), or 1992–2010 (the Netherlands), ii) they had at least 70% of daily data in each month, and iii) they had both $PM_{10}$ and $PM_{2.5}$ data for 2010. An overview of the stations is given in Table S3. Regarding the data treatment, the monthly average was calculated initially for each station separately and the aerosol coarse fraction was subtracted to estimate $PM_{2.5}$ concentrations. Then, an average over the stations was taken before the annual (or summer) average was calculated requiring all 12 (or 3) months being
available for a year to be considered in the analysis. The slope of the Theil-Sen trend gave the absolute concentration change per year ($\mu g\ m^{-3}\ yr^{-1}$), which was then multiplied by the number of year intervals (number of years - 1) to yield the total absolute change for the respective period. The total relative change was estimated by dividing the total absolute change with the regression value of the respective period's initial year.

The changes in the measured $PM_{10}$ and estimated $PM_{2.5}$ concentrations at selected stations over the studied period and the
results of the trend analysis are shown in Fig. 1 and Table 2, respectively, for summer as well as for the whole year. A steeper decreasing trend in $PM_{10}$ concentrations is evident for the Netherlands (-0.92 ± 0.11 $\mu g\ m^{-3}\ yr^{-1}$) compared to Switzerland (-0.64 ± 0.08 $\mu g\ m^{-3}\ yr^{-1}$), especially in the summer (-1.04 ± 0.14 and -0.56 ± 0.08 $\mu g\ m^{-3}\ yr^{-1}$, respectively). The annual total relative change in $PM_{10}$ concentrations is -43% for the Netherlands and -41% for Switzerland. This is in line with the -44% $PM_{10}$ change in Europe for the time period 1992–2009 that was reported by Wang et al. (2012a). Our $PM_{10}$
trend results for Switzerland are also in line with the results (-0.53 and -0.58 $\mu g\ m^{-3}\ yr^{-1}$, for annual and summer respectively) reported by Barmpadimos et al. (2011) for the time period 1991–2008; small differences in the trends between the studies are attributed to the inclusion of more sites (with available data later than 1990) in Barmpadimos et al. (2011).

## 4. Results and discussion

### 4.1 Model evaluation

The model performance evaluation for both WRF and CAMx models was carried out and discussed in detail in Oikonomakis et al. (2017). A summary of the statistical metrics and model performance evaluation is given in Tables 3 and 4, respectively, for the daily mean $O_3$, $PM_{2.5}$ and $PM_{10}$. The model performance for $O_3$ and $PM_{2.5}$ was satisfactory as discussed in detail by Oikonomakis et al. (2017). On the other hand, there was a consistent underestimation of $PM_{10}$ with a mean bias (MB) of -6 $\mu g\ m^{-3}$ and normalized mean bias (NMB) of -33%. However, the correlation coefficient for the $PM_{10}$ is 0.5, suggesting that
the model can capture the observed $PM_{10}$ temporal evolution. Also, since the model performance for $PM_{2.5}$ is better, this implies that the discrepancy in the $PM_{10}$ is more likely due to missing emissions in the coarse fraction such as sea salt, mineral dust and wild fires (see Sect. 2.1). Even with the inclusion of such emissions, models still have difficulties



simulating the $PM_{10}$ concentrations accurately as the uncertainties related to these emissions are large and meteorological uncertainties (e.g. in wind speed, vertical mixing) also play an important role (Karamchandani et al., 2017; Solazzo et al., 2017).

The systematic model underestimation of the $PM_{10}$ concentrations is also evident in the AOD (Table 4), where the model
consistently underestimates the AERONET observations (MB = -0.15, MGE = 0.16). Despite this systematic negative bias, the model is able to represent quite accurately the spatial and temporal variability of the observed AOD, indicated by the relatively high correlation ($r = 0.6$) between the model and the observations. Other possible error sources for the modeled AOD could be: i) the simplified treatment of the aerosol size distribution, the optical properties, and the mixing state (Curci et al., 2015), ii) the use of the constant climatological aerosol Elterman (1968) profile for the upper troposphere and
stratosphere, iii) uncertainties in RH and f(RH) for inorganic aerosols, and/or iii) uncertainties due to grid resolution (horizontal or vertical). Overall, our model AOD discrepancies are within range with other modeling studies (Cesnulyte et al., 2014; Im et al., 2015), where they underline the importance of dust and sea salt treatment in the models.

In the case of SSR, the model performance is better, with a slight overestimation (NMB = 6%; Table 4). The diurnal and inter-daily variability was captured as well (Fig. 2; $r = 0.8$). In general, the overestimation of the downward shortwave
radiation is a long-standing issue in the models (Wild, 2008; Wild et al., 2013), which indicates that it might be related not only to aerosols but also to other important sources of uncertainty such as parameters related to clouds and water vapor. Since the modeling framework of this study is based on the $PM_{2.5}$ we believe that the systematic $PM_{10}$ model bias would not affect the results and conclusions significantly.

## 4.2 PM species

The modeled daytime (10:00–18:00 LMT) concentrations of the fine PM species to be adjusted for AOD and SSR calculations are shown in Fig. 3. Sulfate concentrations were predicted to be the highest in summer among all seven species (Fig. 3a) especially over the Mediterranean Sea and southeastern Europe. Although ship emissions are considered to be the main source of elevated sulfate concentrations over the sea, their contribution to the land areas in Southeastern Europe (e.g. Greece and Turkey) is much smaller compared to other emissions sources, such as power generation, industries and road
transport (Tagaris et al., 2015; Aksoyoglu et al., 2016). Particulate nitrate concentrations, on the other hand, are higher in regions with high $NO_x$ and $NH_3$ emissions (around the English Channel, Benelux region, northern Italy). The concentrations of anthropogenic SOA (Fig. 3d) are very low and the spatial distribution of primary species POA, EC, and FPRM is similar to their emission patterns (Fig. 3e-g). The high POA concentrations on the eastern boundary of the model domain are consistent with the summer 2010 Russian wildfires, which influenced mainly the areas around Moscow and to a lesser extent the
eastern part of Europe (Mei et al., 2011; Portin et al., 2012; Péré et al., 2015). It is noted that, although wild fire emissions are not included in the model (see Sect. 2.1), they enter the model domain from the model boundaries.





### 4.3 Results of PM adjustment scenarios

#### 4.3.1 Changes in AOD

In this section, the AOD in the base case (BASE) is compared to the AOD after the adjustment of fine PM species to represent the conditions in 1990 (see Table 1 for the adjustment scenarios). The simulated AOD in the base case (Fig. 4a)

has a similar spatial distribution over the European domain as the anthropogenic aerosols (see Fig. 3h), although the highest AOD values in the whole grid are in the dust-enriched northwest Africa (in the model dust is included only in the boundary conditions). The European (i.e., excluding northwest Africa) land (land and marine) grid mean of the AOD is 0.14 (0.13), while in more polluted regions (e.g. Po Valley, Benelux region, western Turkey) the AOD values are as high as 0.20-0.25. The spatial distribution of modeled AOD is in line with modeling results and satellite observations (at around 550 nm) from

other studies for different summer periods (Real and Sartelet, 2011; Xing et al., 2015b; Mailler et al., 2016), as well as with the eight-model ensemble results of the $PM_{10}$ spatial distribution for 2010 by Colette et al. (2017).

The changes in the calculated AOD after the adjustment of the PM species according to the descriptions given in Table 1 are shown in Figs. 4b-d. The largest difference in AOD was obtained with the PHOT2 scenario (Fig. 4c) of up to -0.41, followed by PHOT3 (Fig. 4d) and PHOT1 scenarios (Fig. 4b) with up to -0.33 and -0.21, respectively. The continental European grid

averages for the AOD differences between the base case (BASE) and PHOT1, PHOT2 and PHOT3 scenarios are -0.10, -0.21 and -0.15, respectively. The changes in AOD in all three tests consistently follow the spatial distribution of anthropogenic aerosols (see Fig. 3h), with southwestern and northern Europe having the smallest values due to higher contribution of dust and BSOA, respectively, to aerosol concentrations in these regions (Fig. S1). The spatial distribution of the simulated AOD differences (Figs. 4b-d) is similar to that from the modeled difference in $PM_{10}$ concentrations between 1990 and 2010

(Colette et al., 2017), supporting the assumptions used in our sensitivity tests. Xing et al. (2015b) reported that the simulated trends of AOD (at 533 nm) in summer in Europe were -0.007 and -0.003 $yr^{-1}$, for the periods 1990–2000 and 2000–2010 respectively, resulting in -0.1 for the whole period 1990–2010. They also calculated an AOD summer trend of -0.002 to -0.007 $yr^{-1}$ from the analysis of satellite observations for the period of 2000–2010. Turnock et al. (2015) reported modeled and observed (from AERONET sites) summer AOD (at 440 nm) trends of -0.005 and -0.014 $yr^{-1}$ respectively for the period

2000–2009, which are higher than the ones reported by Xing et al. (2015b) probably due to the lower wavelength used by Turnock et al. (2015). This could be an indication that the fine-mode particles were mainly responsible for the decreasing AOD trends as their scattering efficiency is higher at smaller wavelengths (Seinfeld and Pandis, 2016). Li et al. (2014a) also suggested that the AOD reduction in Europe might have been driven by decreases in the fine-mode particles. The authors reported decreasing trends in the AOD (at 440 nm), as well as in the Ångström exponent (at 440/870 nm), for the vast

majority of the European AERONET sites between 2000–2013; the largest AOD decrease was observed in western Europe with -0.1 $decade^{-1}$ (i.e., -0.010 $yr^{-1}$). Another study by Bin et al. (2017) further supported the conclusions about the AOD decreasing due to the smaller particles. They showed that the AOD (at 555 nm) trend from satellite observations for western Europe in summer was ~ -0.003 $yr^{-1}$ between 2001 and 2015. Assuming that the AOD trend between 1990–2000 and 2000–



2010 was the same, we estimated the AOD trend in Europe for 1990–2010 to be ~ -0.005, -0.010 and -0.008 yr$^{-1}$ for the PHOT1, PHOT2 and PHOT3 scenarios, respectively. Our results about the change in AOD are in the same range as the other studies, by taking into account that: i) for smaller wavelengths (350 nm in our case and > 440 nm in the aforementioned studies) larger changes are expected due to the higher decreasing trend in the fine-mode particle concentrations as discussed

above, ii) the AOD reduction might have been larger for 1990–2000 than 2000–2010 (Xing et al., 2015b).

### 4.3.2 Changes in SSR

In this section, the SSR in the base case (BASE) is compared to the SSR after the adjustment of fine PM species to represent the conditions in 1990 (see Table 1 for the adjustment scenarios). The modeled SSR for the base case (BASE) is shown in Fig. 5a. The model captured both the magnitude and the spatial distribution with the south-north latitudinal gradient and the

lowest values over the northwest Atlantic Ocean, as also shown by other studies (Forkel et al., 2012, 2015). The average (maximum) differences in SSR over land between the base case (BASE) and PHOT1, PHOT2 and PHOT3 tests are 9 (20), 17 (35) and 11 (26) W m$^{-2}$, respectively (Fig. 5b-d). Following the same method as for the AOD, we estimated the SSR trend as 0.45, 0.85 and 0.55 W m$^{-2}$ yr$^{-1}$ for PHOT1, PHOT2 and PHOT3, respectively. Other studies reported modeled and observed SSR trends within a range of 0.35–0.55 W m$^{-2}$ yr$^{-1}$ for different periods between 1986 and 2012 (Norris and Wild,

2007; Allen et al., 2013; Cherian et al., 2014; Nabat et al., 2014; Sanchez-Lorenzo et al., 2015; Turnock et al., 2015). Based on these studies, PHOT1 and PHOT3 are more realistic scenarios than PHOT2 which seems to present a slight overestimation of the DRE changes. Xing et al. (2015a) reported for Europe SSR changes between 1990 and 2010 in the range of 6-18 W m$^{-2}$ in line with PHOT1 and PHOT3 scenarios (Fig. 5b-d). In general, our simulated AOD and SSR changes between 1990 and 2010 for PHOT1 and PHOT3 scenarios seem to be consistent with respective observed and modeled

changes from other studies, while PHOT2 scenario can be considered rather an upper limit of the DRE changes.

### 4.4 Effects on ozone via photolysis rates

The simulated (in the base case) photolysis rate of $NO_2$, $J(NO_2)$, consistently follows the south-to-north latitudinal gradient of SSR and temperature, as shown in Fig. 6a. The modeled continental mean absolute (relative) differences in $J(NO_2)$ between the base case (BASE) and PHOT1, PHOT2 and PHOT3 tests are 0.7 (3%), 1.3 (6%) and 0.9 (4%) hr$^{-1}$, respectively

(Figs. 6b-d). The spatial distribution and relative changes are the same for the photolysis rate of $O_3$, $J(O_3 \rightarrow O^1D)$, with changes in absolute terms being 0.0015, 0.0029 and 0.0020 hr$^{-1}$ respectively for PHOT1, PHOT2 and PHOT3 tests (Fig. S2). As discussed in Sect. 1, changes in the photolysis rates will affect the chemical production and destruction of ozone as well as other chemical processes in the troposphere such as the secondary aerosol (SA) formation, which in turn can affect back the photolysis rates. This implication, however, is rather small with the change in SA concentration (continental grid mean)

between base case (BASE) and PHOT1, PHOT2 scenarios being 0.01 and 0.02 μg m$^{-3}$ respectively (Figs. S3–S4), or in relative terms 0.3% and 0.6% respectively (the respective results for the PHOT3 test are very similar to the ones of the



PHOT1 test and are therefore not shown). We conclude that these changes in SA have negligible impact on the photolysis rates.

The enhancement of the photolysis rates leads to higher ozone formation especially in regions where there are significant ozone precursor emissions (i.e., central Europe, northern Italy; Fig. 7). The magnitude of the effect of enhanced photolysis

rates is, however, rather small on average. The difference in the surface ozone between BASE and PHOT2 scenarios during the daytime (10:00–18:00 LMT) varies between 0.4 and 0.7 ppb (1–1.5%) over central Europe and up to 1.4 ppb (2.5%) in the Po Valley, while it is smaller for the other two scenarios (up to 0.7–0.8 ppb, 0.7–1.4%). However, on an hourly resolution the largest difference in surface ozone between BASE and PHOT2 scenarios can go up to 8 ppb (16%) and up to 4 ppb (10%) for the PHOT1 and PHOT3 scenarios in high-$NO_x$ areas on land (Fig. S5).

We repeated similar tests based on a second base case (BASE_$NO_x$) with increased $NO_x$ emissions which improved the model performance for ozone production as discussed in Oikonomakis et al (2017). In the BASE_$NO_x$ case, ozone production was higher over a larger area compared to the BASE (see Figs. 7a and 8a). Consequently, the difference in ozone between BASE_$NO_x$ and the scenarios PHOT1_$NO_x$, PHOT2_$NO_x$ and PHOT3_$NO_x$ were more pronounced over a larger area, the magnitude of the impact, however, only slightly increased (Figs. 8 and S5).

## 4.5. Effects on ozone via BVOCs emissions

The response of isoprene emissions (2.5-3% changes) to SSR changes (3%) is nearly linear (Fig. 9), in line with the literature (Guenther et al., 2006). On the contrary, terpene (monoterpene and sesquiterpene) emissions are less sensitive to SSR with changes up to 0.7% (Fig. 9). Nevertheless, the BVOCs emissions sensitivity to solar radiation can vary depending on the model parameterization of physical processes such as the emission dependence on light and the canopy calculations of

diffuse and direct radiation as well as the relative contribution between shaded and sunlit leaves over multiple leaf area index (LAI) layers (Messina et al., 2016). In general, BVOCs emission estimates have high uncertainties (a factor of 2–3) due to uncertainties in the land use, LAI and parameterization of physical processes, the large number of compounds and biological sources, and the lack of observations (Guenther et al., 2006, 2013; Karl et al., 2009; Oderbolz et al., 2013). Despite these uncertainties, Stavrakou et al. (2014) also reported a linear response of isoprene emissions with the respective SSR changes

in Asia between 1979 and 2012, using a different biogenic emission model. On the other hand, other studies suggested that the photosynthetically active radiation (PAR), which depends more on the diffuse component of solar radiation, did not have a significant impact on the increasing BVOCs trends in Europe during the solar "brightening" (after 1980) probably due to the diffuse to direct radiation ratio decrease, compensating for the total increase in SSR (Mercado et al., 2009; Yue et al., 2015). In fact, during the solar "dimming" (i.e., when the total SSR decreased) between 1960 and 1980 both the diffuse

fraction of PAR and the photosynthesis were enhanced (Mercado et al., 2009). It is further suggested that the BVOCs emissions are less sensitive to the SSR compared to the temperature, which is identified as a more important driver for the BVOCs emission trends (Guenther et al., 2006; Lathière et al., 2006; Yue et al., 2015; Gustafson et al., 2017).





The impact of a 2.5-3% and 0.7% increase in isoprene and terpene emissions (BIO scenario), respectively, on daytime (10:00–18:00 LMT) average surface ozone is rather small (up to 0.08 ppb, ~ 0.2%; Fig. 10a) and an order of magnitude smaller than the respective ozone impact via photolysis rates (see Fig. 8). Both the daytime average and largest hourly (~1 ppb) impacts are higher in central Europe where both BVOCs and $NO_x$ emissions are ample (Figs. 10a and S6). The effects

of increased BVOCs emissions are higher in magnitude (up to 0.11 ppb, ~ 0.3%) and spatial coverage when applied to the base case with higher $NO_x$ emissions (i.e., $BASE\_NO_x – BIO\_NO_x$), as shown in Figs. 10b and S6. The combined effects via BVOCs emissions and photolysis rates (COMBO and $COMBO\_NO_x$ scenarios) on surface ozone appear to be roughly additive, with the photolysis rates effects dominating the overall impact (daytime average difference was up to 0.8 ppb, 1.5%; Figs. 10c-d and S6). Overall, the direct effects of SSR changes on the BVOCs emissions (with the assumptions and

parameterizations of this study) were small, and as a result this was also the case for the consequent impact on surface ozone. However, SSR trends implications related to temperature and $CO_2$ changes (Wild et al., 2007; Storelvmo et al., 2016) might have a more significant impact on BVOCs emissions and thus on surface ozone, but this was beyond the scope of this study.

### 4.6 Solar "brightening" and ozone trends

Although the effects of DRE changes via photolysis rates and BVOCs emissions on surface ozone seem to be small

compared to the total ozone concentrations, it might be more meaningful to compare with the magnitude of the observed ozone concentration trends. Wilson et al. (2012) reported an annual (summer) increasing trend of 0.16 ± 0.02 (0.12 ± 0.06) ppb yr[-1] in the European ground-level ozone (stations-average) for the period 1996-2005, while Colette et al. (2016) reported an annual ozone trend (European Monitoring and Evaluation Programme (EMEP) network-median) of 0.06 ppb yr[-1] between 1990 and 2012. The total ozone difference (0.2–0.8 ppb) via both the effects on photolysis rates and BVOCs emissions

(COMBO scenario) would translate (considering the full 20-year time period) to a summer trend of 0.01–0.04 ppb yr[-1]. Although these values should not be considered for a direct comparison with the absolute values of the aforementioned observed ozone trends (due to differences in the data analysis like time averaging and spatial coverage), the comparison of their order of magnitude suggests a higher importance of the impact of solar "brightening" on surface ozone than when just comparing to the total ozone concentrations. Therefore, the aforementioned comparison indicates that the solar "brightening"

might have had an accountable impact on the European surface ozone trends since the 1990s and could have partially dampened the effects of ozone precursor emissions reduction along with other more influential physical processes like intercontinental transport and stratosphere–troposphere exchange (Ordóñez et al., 2007; Derwent et al., 2008, 2015).

### 5. Conclusions

We investigated the impact of solar "brightening" on European summer surface ozone between 1990 and 2010 using the

CAMx air quality model. We modeled the summer of 2010 as base case and designed various sensitivity tests based on literature review as well as an observational PM trend analysis performed in this study to represent the AOD and SSR




conditions of the year 1990. One of the main assumptions in this study was that the change in DRE was the main driver for the solar "brightening" in Europe and thus excluding the AIE and cloud cover natural variability. Moreover, this study focused on the less uncertain effects of DRE via the impact on photolysis rates and BVOCs emissions, compared to the more uncertain DRE-induced meteorological effects. Lastly, in the model scenarios we assumed that the AOD changes between

1990 and 2010 in Europe were predominantly driven by changes in the anthropogenic $PM_{2.5}$ concentrations, and hence we excluded any AOD changes due to variations in $PM_{10}$ or natural $PM_{2.5}$ concentrations.

Regarding the impact on ozone via photolysis rates, the PHOT1 and PHOT3 model scenarios (doubling anthropogenic $PM_{2.5}$ concentrations and increasing only sulfate concentrations by 3.4 times, respectively) were considered to be closer to the observed and modeled AOD and SSR changes reported by other studies (see Sections 4.3.1 and 4.3.2) compared to PHOT2

scenario (tripling anthropogenic $PM_{2.5}$ concentrations) that should be regarded as an upper limit. Furthermore, the PHOT3 scenario was based on less uncertain assumptions (well-documented sulfate concentrations trends; see Sections 2.3.1 and 3) and therefore we considered it to be more realistic (except for southeastern Europe where the effects might be overestimated). The differences in AOD, SSR and the main ground-level photolysis rates ($J(NO_2)$ and $J(O_3 \rightarrow O^1D)$) between the BASE and PHOT3 scenarios (representing the changes between summer of 1990 and 2010) were -0.33, 11 W

$m^{-2}$ and 4%, respectively and the consequent impact on daytime (10:00–18:00 LMT) surface ozone was on average 0.2–0.4 ppb (0.5-1%) over central and western Europe. Moreover, the largest hourly difference in surface ozone could be as high as 4 ppb (10%), while the same test performed on a base case with higher $NO_x$ emissions and ozone production (BASE_ $NO_x$ – PHOT3_ $NO_x$) resulted in an extension of the spatial coverage of the DRE on ozone (apart from the VOC-limited Benelux region).

On the other hand, the impact of -3% SSR change resulted in a near-linear response in isoprene emissions (2.5-3%) but less in the terpene (monoterpene and sesquiterpene) emissions (0.7%), with the subsequent effects on daytime ozone being small (up to 0.08 ppb, ~ 0.2%). Compared to the impact on ozone via the photolysis rates, the effects of BVOCs emission changes were about an order of magnitude smaller and thus the former dominated the latter impact when they were combined, as their effects were nearly additive. Therefore the overall impact of SSR changes on ozone remained relatively small.

Nevertheless, the role of the solar "brightening" (as quantified in this study) in the European summer surface ozone trends was suggested to be more important when comparing to the order of magnitude of the ozone trends instead of the total ozone concentrations.

Finally, the inclusion of the impact of DRE on meteorology and AIE might have additional increasing or, conversely, decreasing effects on surface ozone as discussed in Sect. 1. However, climate modeling studies show that the decline of

aerosols can also affect the global atmospheric circulation as well as the atmospheric stability (Rotstayn et al., 2014; Wang et al., 2016; Navarro et al., 2017) and this entanglement might have compelling implications on air quality at a regional scale. It is therefore suggested that future air quality studies take into account the possible repercussions of declining aerosols on climate and atmospheric circulation at a global scale for a better understanding of the anthropogenic influence on air quality and climate as well as their complex interlinkage.





*Competing interests*. The authors declare that they have no conflict of interest.

*Acknowledgments*. We would like to thank the following agencies for preparing the datasets used in this study: TNO for the anthropogenic emission inventory; the European Environmental Agency (EEA) and the Swiss National Air Pollution Monitoring Network (NABEL) for the air quality data; the European Centre for Medium-Range Weather Forecasts (ECMWF) and Baseline Surface Radiation Network (BSRN) for the meteorological data; the National Aeronautics and Space Administration (NASA) and its data-contributing agencies (NCAR, UCAR, AERONET) for the TOMS, MODIS and AOD data, the global air quality model data and the TUV model. Calculations of meteorological data were performed with the Swiss National Supercomputing Centre (CSCS). Our thanks extend to RAMBOLL ENVIRON and especially Cristopher Emery for their continuous support of the CAMx model. This work was financially supported by the Swiss Federal Office of Environment (FOEN).

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



**Tables**

**Table 1.** Summary of model runs. All runs used the emissions and meteorology of 2010.

| Scenario | Description |
| --- | --- |
| BASE | Base case using the default parameterization as described in Section 2.1. |
| BASE_$NO_x$ | Same parameterization as BASE scenario, but with doubled $NO_x$ emissions to be used as a second base case with higher ozone production according to Oikonomakis et al. (2017) . |
| PHOT1 | Increased concentrations of $SO_4^{-2}$, $NH_4^+$, $NO_3^-$, POA, ASOA, EC, FPRM by a factor of 2 over land only in the calculation of AOD. |
| PHOT1_$NO_x$ | Same method as PHOT1 but applied on the BASE_$NO_x$ scenario. |
| PHOT2 | Increased concentrations of $SO_4^{-2}$, $NH_4^+$, $NO_3^-$, POA, ASOA, EC, FPRM by a factor of 3 over land only in the calculation of AOD. |
| PHOT2_$NO_x$ | Same method as PHOT2 but applied on the BASE_$NO_x$ scenario. |
| PHOT3 | Increased concentrations of only $SO_4^{-2}$ by a factor of 3.4 and only in the calculation of AOD. |
| PHOT3_$NO_x$ | Same method as PHOT3 but applied on the BASE_$NO_x$ scenario. |
| BIO | Re-run of the BASE scenario with new biogenic emissions generated after decreasing SSR by 3% in the biogenic emission model. |
| BIO_$NO_x$ | Same method as BIO but applied on the BASE_$NO_x$ scenario. |



COMBO          A combination of the PHOT3 and BIO scenarios.

COMBO_NO$_x$    A combination of the PHOT3_NO$_x$ and BIO_NO$_x$ scenarios.

**Table 2.** Trends (and their standard errors) and total changes in PM$_{10}$ concentrations measured at 3 stations in Switzerland (1990–2010) and at 3 stations in the Netherlands (1992–2010). The total relative changes in the estimated PM$_{2.5}$ concentrations are also reported in parentheses. All trends are statistically significant (at the 99% confidence level).

|  | Trend (µg m$^{-3}$ yr$^{-1}$) | | Absolute change (µg m$^{-3}$) | | Relative change (%) | |
|---|---|---|---|---|---|---|
|  | Annual | Summer | Annual | Summer | Annual | Summer |
| Switzerland | -0.64 ± 0.08 | -0.56 ± 0.08 | -12.7 | -11.2 | -41 (-48) | -45 (-53) |
| The Netherlands | -0.92 ± 0.11 | -1.04 ± 0.14 | -16.6 | -18.6 | -43 (-55) | -50 (-65) |

**Table 3.** Definition of statistical metrics for model performance evaluation. $M_i$ and $O_i$ stand for modeled and observed values, respectively and $N$ is the total number of paired values.

| Metric | Definition |
|---|---|
| Mean Bias (MB) | $MB = \dfrac{1}{N}\sum_{i=1}^{N}(M_i - O_i)$ |
| Mean Gross Error (MGE) | $MGE = \dfrac{1}{N}\sum_{i=1}^{N}|M_i - O_i|$ |
| Root-Mean-Square Error (RMSE) | $RMSE = \sqrt{\dfrac{1}{N}\sum_{i=1}^{N}(M_i - O_i)^2}$ |
| Normalized Mean Bias (NMB) | $NMB = \dfrac{\sum_{i=1}^{N}M_i - O_i}{\sum_{i=1}^{N}O_i}$ |
| Normalized Mean Error (NME) | $NME = \dfrac{\sum_{i=1}^{N}|M_i - O_i|}{\sum_{i=1}^{N}O_i}$ |
| Pearson correlation coefficient ($r$) | $r = \dfrac{\sum_{i=1}^{N}(M_i - \bar{M}) \cdot (O_i - \bar{O})}{\sqrt{\sum_{i=1}^{N}(M_i - \bar{M})^2} \cdot \sqrt{\sum_{i=1}^{N}(O_i - \bar{O})^2}}$ |





**Table 4.** Statistical summary of model performance evaluation for summer 2010. The units for MB, MGE and RMSE are in ppb for $O_3$, in µg m$^{-3}$ for the PM and in W m$^{-2}$ for the SSR, while the units for NMB and NME are in %.

|  | No. of stations | MB | MGE | RMSE | NMB | NME | $r$ |
|---|---|---|---|---|---|---|---|
| $O_3$ | 347 | 4 | 7 | 8 | 12 | 20 | 0.7 |
| $PM_{2.5}$ | 23 | -0.4 | 5 | 7 | 1 | 44 | 0.5 |
| $PM_{10}$ | 103 | -6 | 8 | 10 | -33 | 45 | 0.5 |
| SSR | 7 | 14 | 35 | 50 | 6 | 15 | 0.8 |
| AOD | 47 | -0.15 | 0.16 | 0.20 | -47 | 51 | 0.6 |





**Figures**

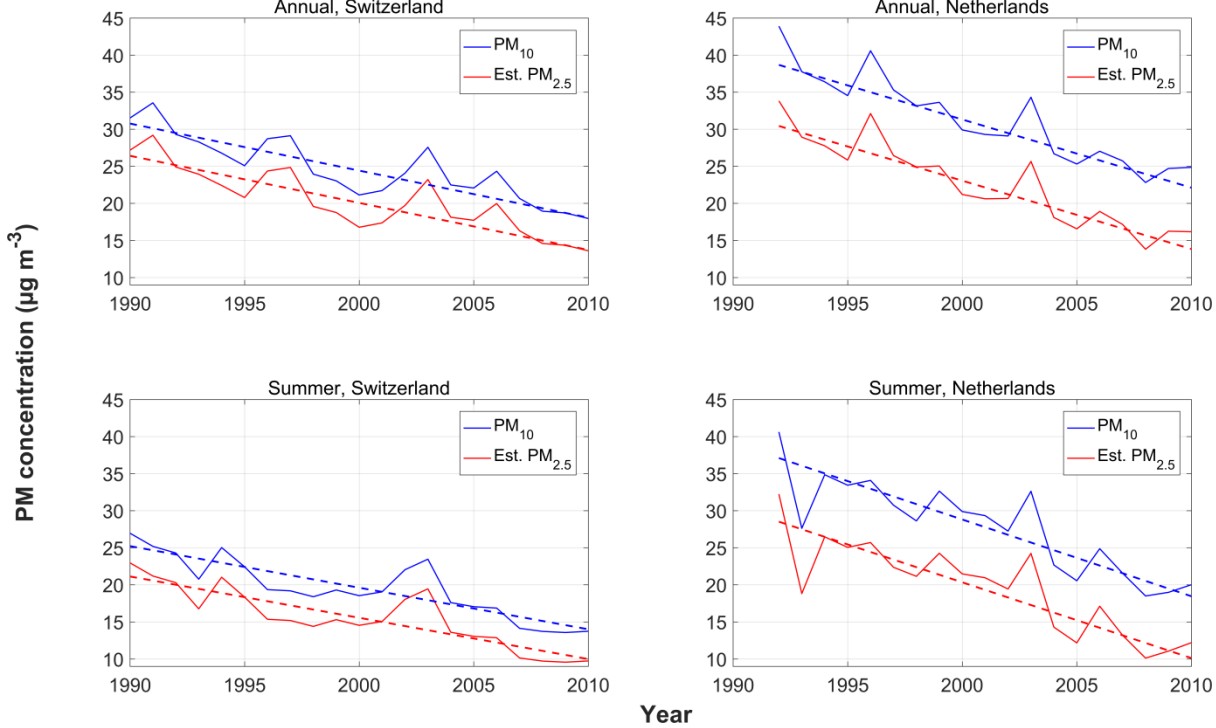

**Figure 1.** Annual (top panels) and summer (bottom panels) concentrations of PM$_{10}$ (blue) and PM$_{2.5}$ (red) measured at 3 stations in Switzerland (left panels) and at 3 stations in the Netherlands (right panels) for the period 1990-2010 and 1992-2010, respectively. Dashed lines show the linear regression fit. PM$_{2.5}$ concentrations were estimated as described in Section 3.





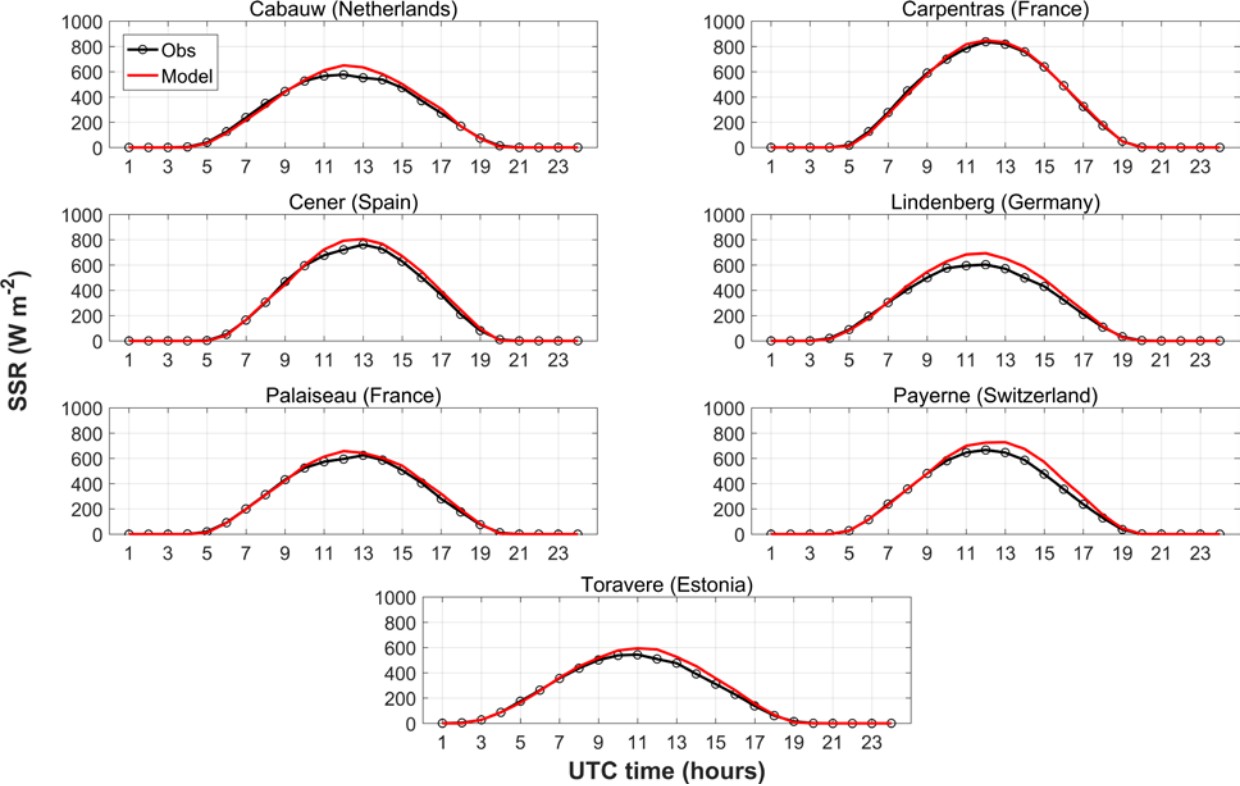

**Figure 2.** Mean diurnal profiles of observed and modeled (BASE scenario) SSR at 7 European sites from the BSRN network in summer 2010.



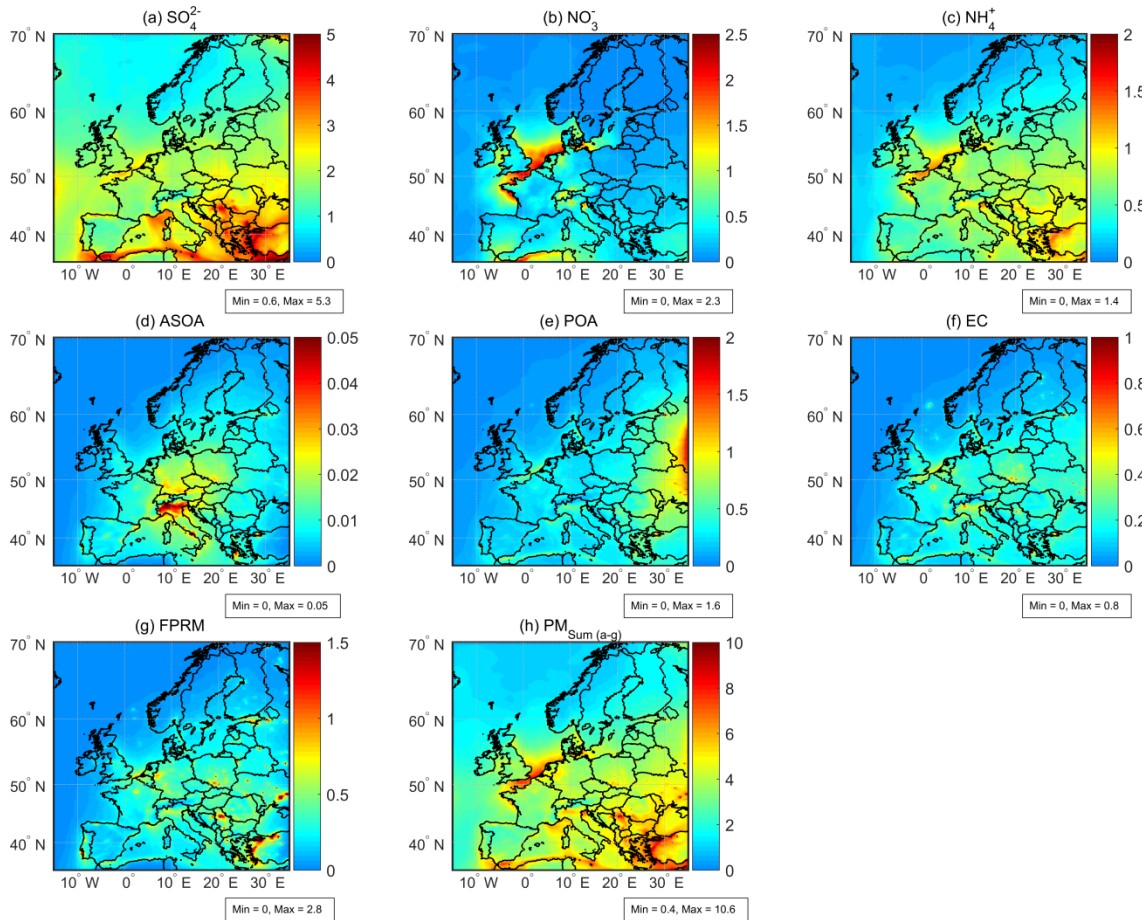

**Figure 3.** Seasonal daytime (10:00-18:00 LMT) mean concentrations (µg m$^{-3}$) of (a) sulfate (SO$_4^{2-}$), (b) nitrate (NO$_3^-$), (c) ammonium (NH$_4^+$), (d) anthropogenic secondary organic aerosol (ASOA), (e) primary organic aerosol (POA), (f) elemental carbon (EC), (g) fine other primary aerosols (FPRM) and (h) sum of a-g, for the BASE scenario in summer 2010.





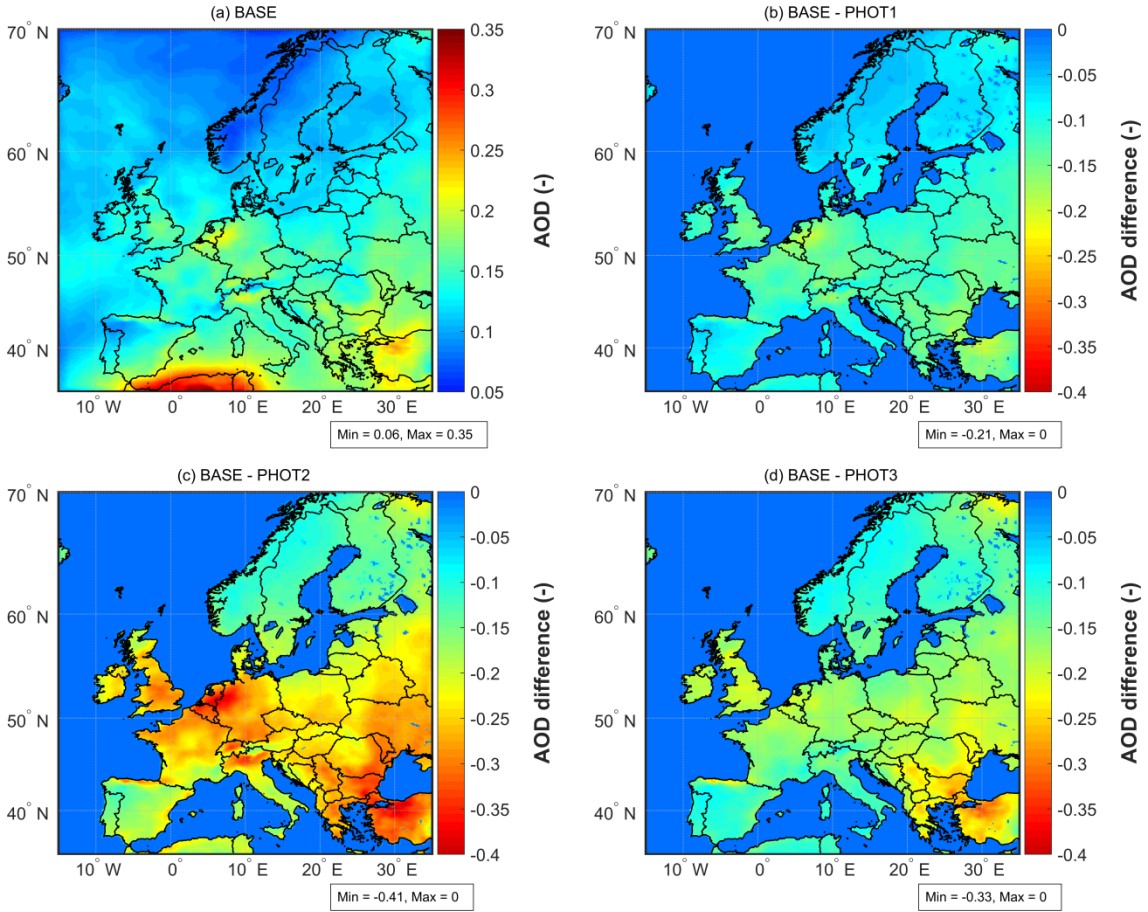

**Figure 4.** Seasonal daytime (10:00-18:00 LMT) mean AOD at 350 nm for the BASE scenario (a) and AOD differences between the BASE scenario and PHOT1, PHOT2 and PHOT3 scenarios (b-d), respectively, in summer 2010. Note the reversed color order in the color scales of panels (b-d).



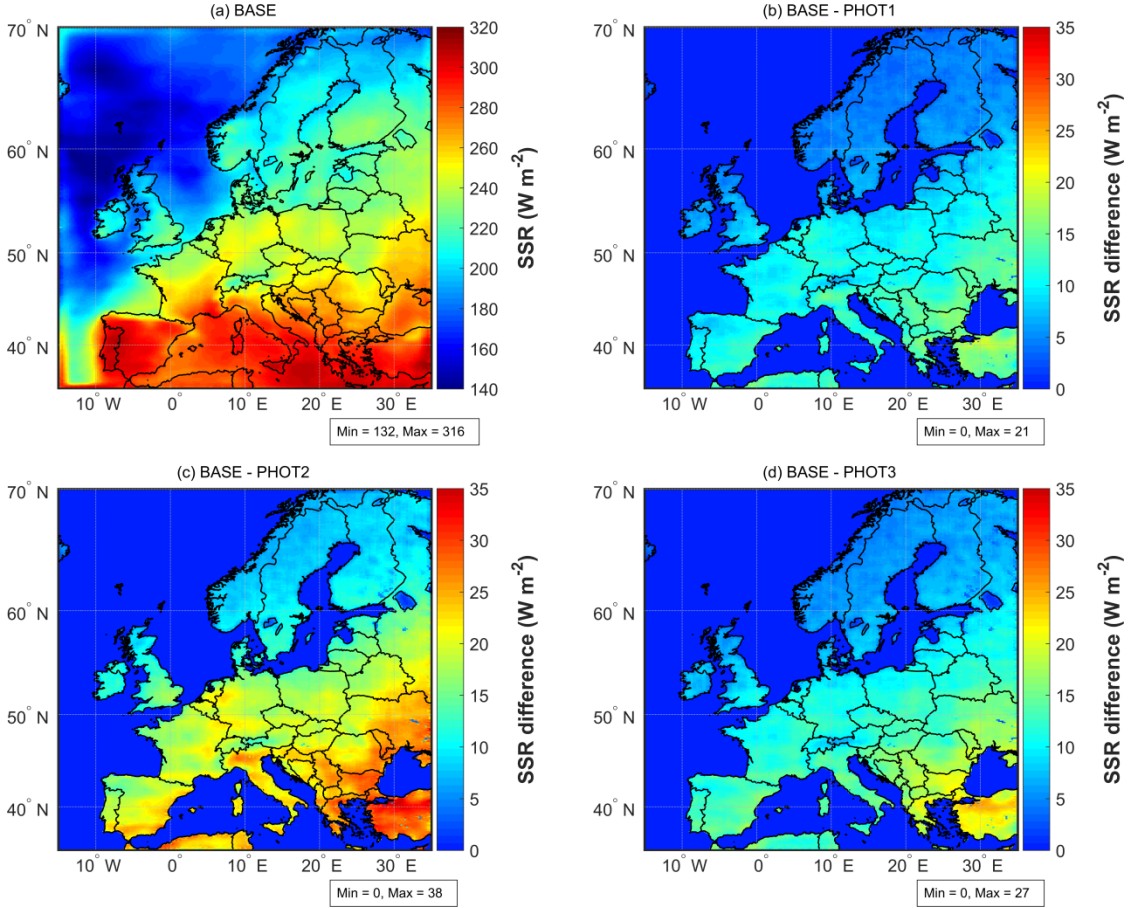

**Figure 5.** Seasonal daily mean SSR for the BASE scenario (a) and SSR differences between the BASE scenario and PHOT1, PHOT2 and PHOT3 scenarios (b-d), respectively, in summer 2010. Note the different color scale between panel (a) and panels (b-d).





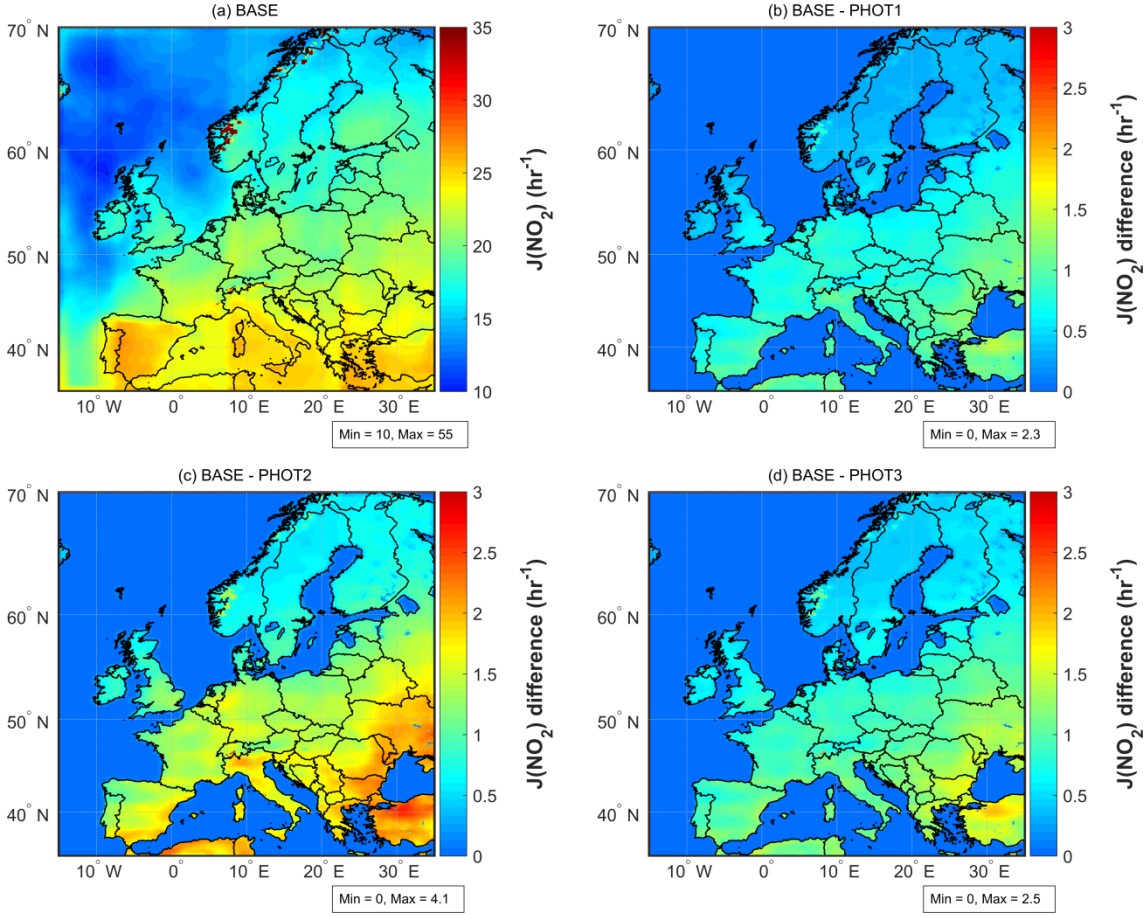

**Figure 6.** Seasonal daytime (10:00-18:00 LMT) mean J(NO$_2$) for the BASE scenario (a) and J(NO$_2$) differences between the BASE scenario and PHOT1, PHOT2 and PHOT3 scenarios (b-d), respectively, in summer 2010. Note the different color scale between panel (a) and panels (b-d).





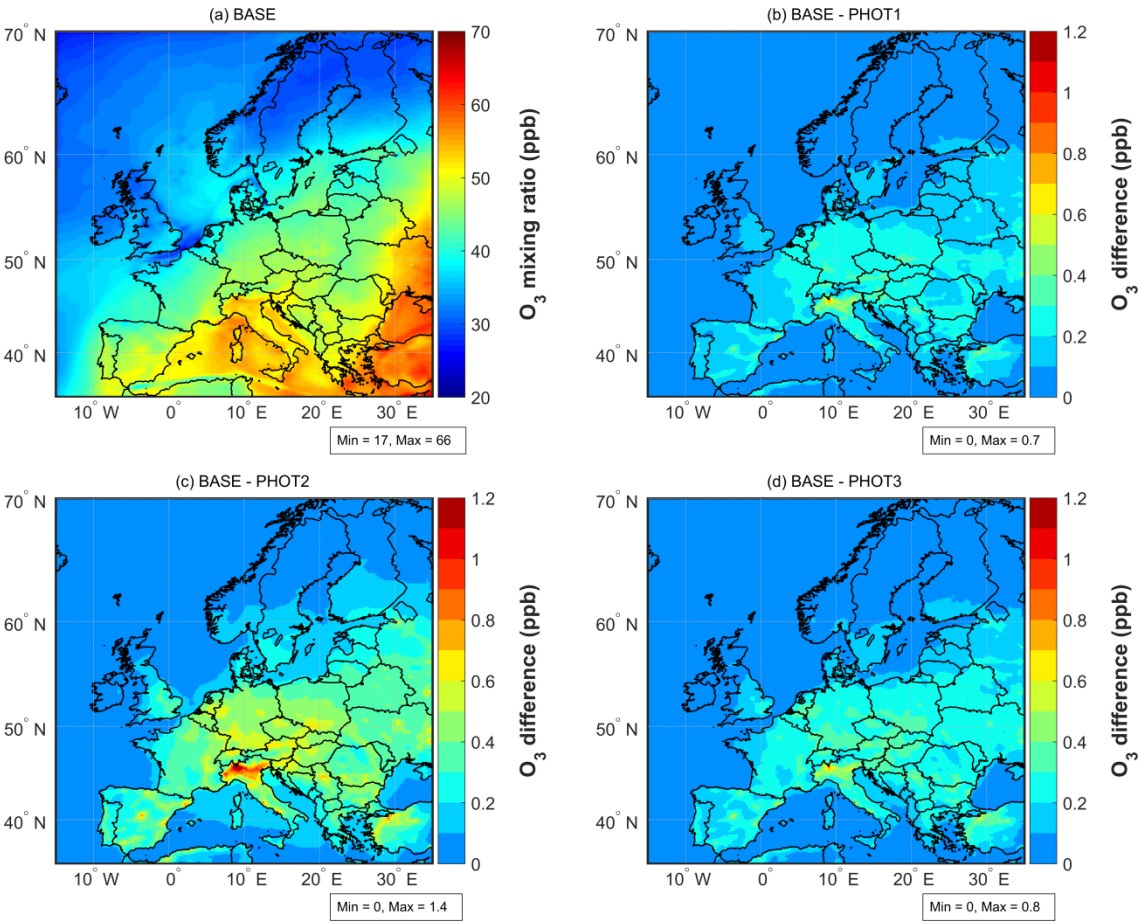

**Figure 7.** Seasonal daytime (10:00-18:00 LMT) mean $O_3$ mixing ratios for the BASE scenario (a) and $O_3$ differences between the BASE scenario and PHOT1, PHOT2 and PHOT3 scenarios (b-d), respectively, in summer 2010. Note the different color scale between panel (a) and panels (b-d).





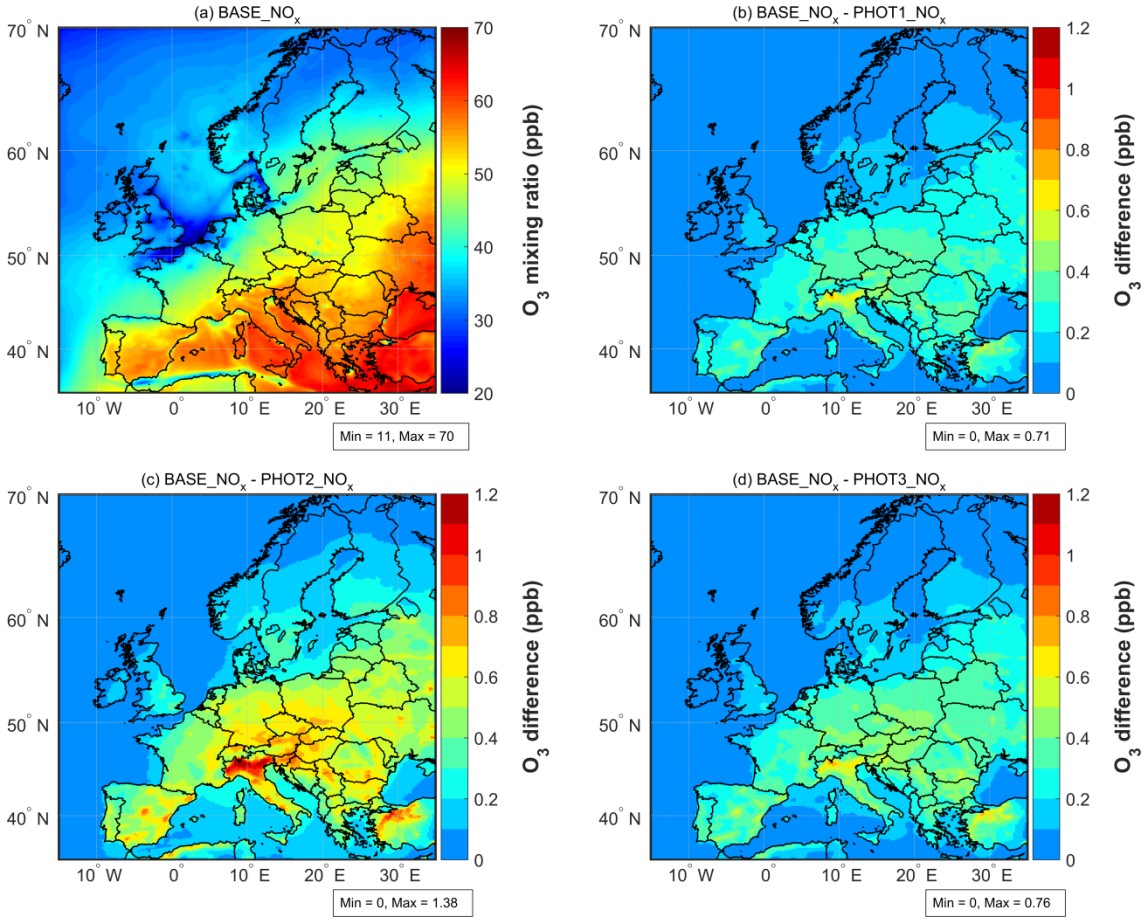

**Figure 8.** Seasonal daytime (10:00-18:00 LMT) mean $O_3$ mixing ratios for the BASE_$NO_x$ scenario (a) and $O_3$ differences between the BASE_$NO_x$ scenario and PHOT1_$NO_x$, PHOT2_$NO_x$ and PHOT3_$NO_x$ scenarios (b-d), respectively, in summer 2010. Note the different color scale between panel (a) and panels (b-d).



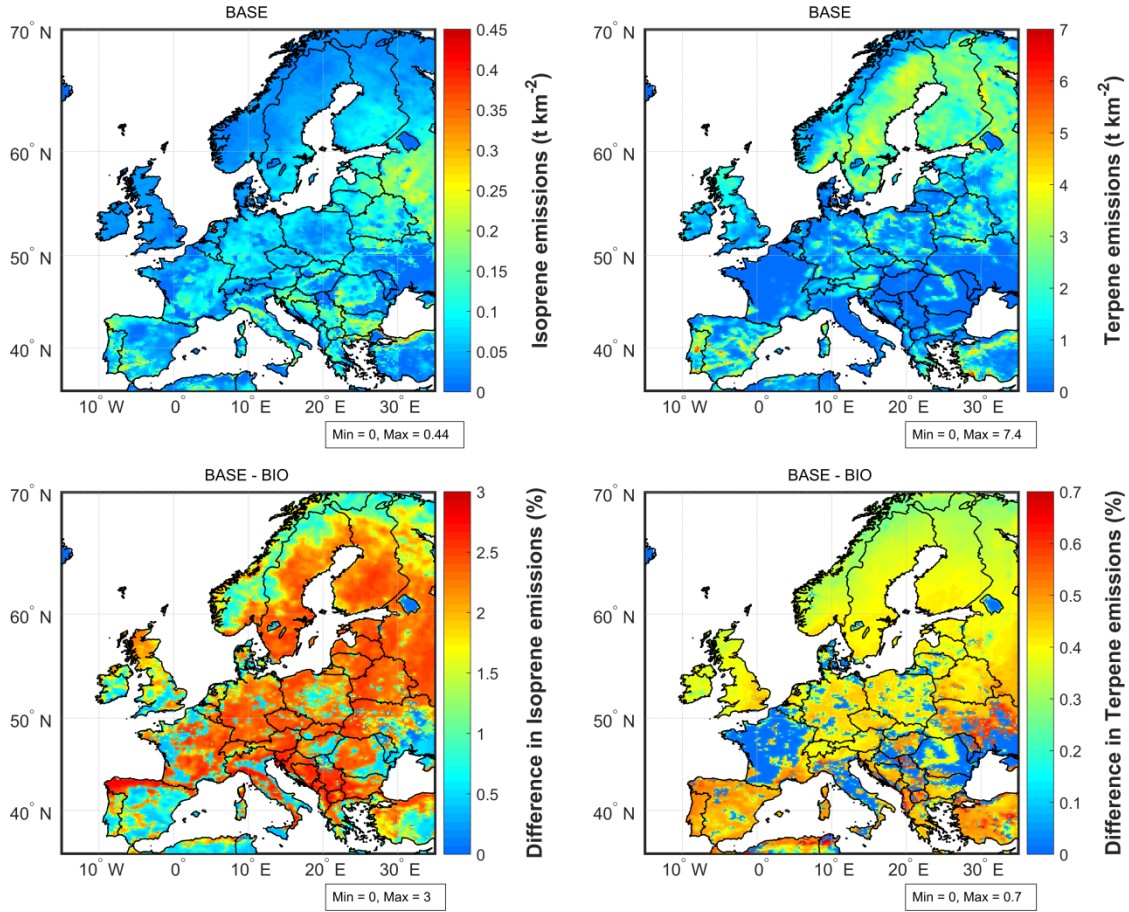

**Figure 9.** Total (i.e. JJA sum) of isoprene (left panels) and terpene (monoterpene and sesquiterpene; right panels) emissions per km² for the BASE scenario (top panels) and relative difference between BASE and BIO scenarios (bottom panels) in summer 2010.





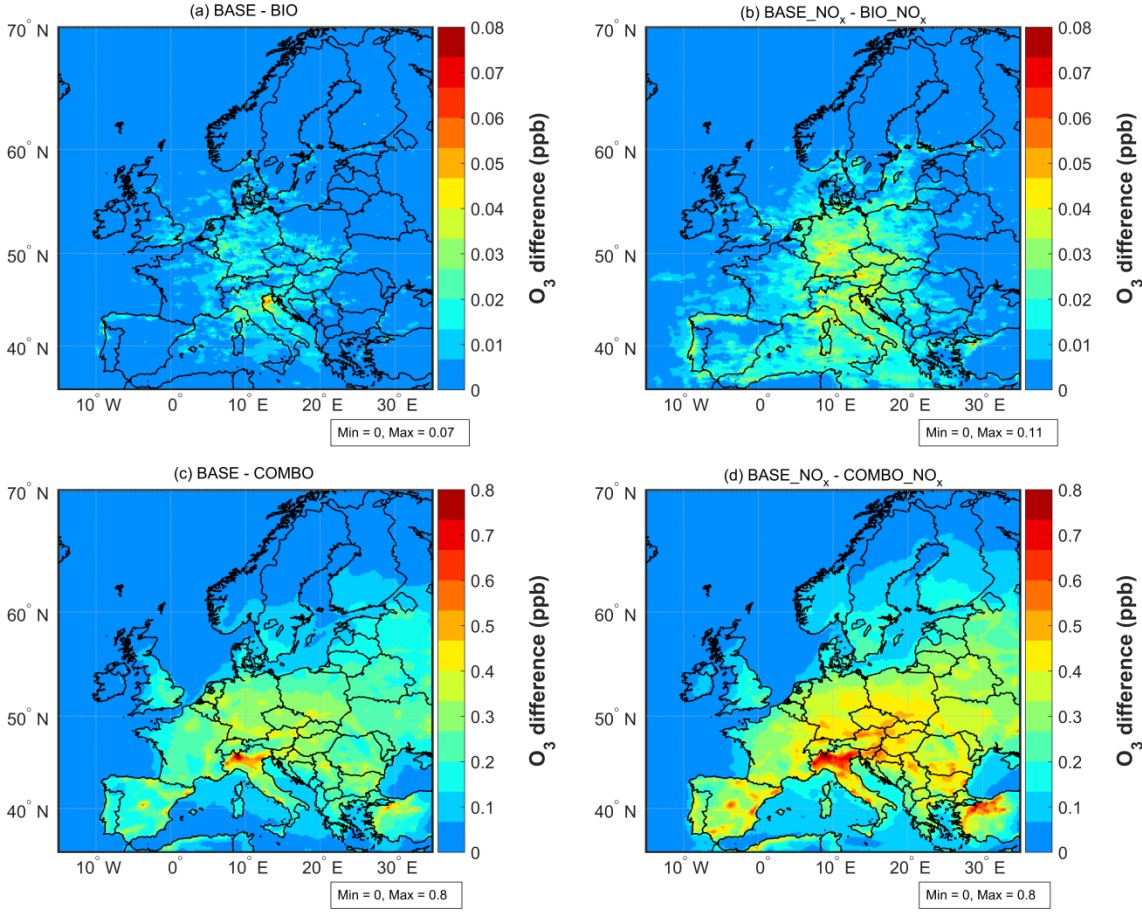

**Figure 10.** Seasonal daytime (10:00-18:00 LMT) mean $O_3$ differences between the (a) BASE and BIO, (b) BASE_NO$_x$ and BIO_NO$_x$, (c) BASE and COMBO, and (d) BASE_NO$_x$ and COMBO_NO$_x$ scenarios in summer 2010.

