# Peer review of "Solar "brightening" impact on summer surface ozone between 1990 and 2010 in Europe – A model sensitivity study of the influence of the aerosol-radiation interactions"

_Atmospheric Chemistry and Physics, 2017_

## Referee Comment (RC1) · Anonymous Referee #1 · 10 Feb 2018

The authors explore the summer surface O3 change over Europe in response to the potential aerosol emission change from 1990 to 2010 via their impact on direct radiation effect (DRE) using CAMx air quality model. They expand their study by taking into account the feedback of biogenic emission through the chain of aerosol emission change to radiation change and perform all studies at different chemistry backgrounds (i.e. base and high NOx emissions). This is an interesting study and the study content is suitable for ACP. I recommend publishing the paper after the authors make minor modifications suggested below.

General Remarks:

[Figure]

It would be good if the authors could explain why it is better to use the designed emissions in 1990 instead of using the emissions in 1990 provided directly from emission inventories. The authors use emission inventory TNO-MACC-III for 2010. To derive the emission in 1990, they first analyze the observed PM10 between 90s and 2010 over 3 Netherlands stations and 3 Switzerland stations to infer potential enhancement factors of emission in 1990 verses in 2010. They then generate three potential emissions representing the situation in 1990 by increasing the emissions in 2010 with these enhancement factors. However, there are emission inventories, such as A2-ACCMIP that can provide emissions directly back to 1980. A2-ACCMIP is one of the multi-year emission datasets available from the international initiative AeroCom project for its second phase (A2) hind-cast model experiments (http://aerocom.met.no) (Chin et al., 2014).

It might also be good to claim the study as a sensitivity study of O3 change in response to aerosol emission change via DRE. The impact of solar "brightening" on surface ozone is not limited to aerosol DRE. Aerosol indirect effect (AIE) is another potential pathway as the authors discussed in the paper. Although the authors indicated that AIE is not a driving reason for surface solar radiation (SSR) trend over Europe during the study period based on some previous studies, AIE is still a non-negligible factor as suggested by a recent study (Mian Chin 2018, personal communication). Furthermore, aerosol can impact ozone chemistry via heterogeneous reaction and the effects of photolysis and heterogeneous update are typically nonlinear (Bian et al., 2003). Feedbacks of O3 production and loss due to changes in O3 precursors leads to the complicated nonlinear feature.

Specific comments 1. Page 1 lines 13-14 in Abstract: PHOT1, PHOT2, and PHO#3 is not in the effects of increased radiation on photolysis rates. 2. Page 2 line 14: $\varphi$ and $\sigma$ depend not only on the gaseous species and air temperature, but also on air pressure for some VOCs. 3. Page 2 line 15: Please add "O2, O3 and" before "water vapor". Please also add references of Bian et al., 2002 and Wild et al., 2000 at the end of this sentence. 4. Page 5 line 9: Is the two-weeks long enough spin-up for O3

chemistry? 5. Page 7 line 14: Could you elaborate on how to increase NOx emission? 6. Page 7 line 22: Do you really use the same emissions for both cases? 7. Page 9 line 29: It seems to me that you are not holding the aerosol coarse fraction constant during the period, but holding the absolute coarse model aerosol amount unchanged. 8. Page 10 lines 4-6: The criteria III should not be listed here since all PM2.5 data in this study are estimated based on PM10, while not measured. 9. Page 11 line 28: What aerosol components are included in FPRM? 10. Page 12 lines 18-20: To support the assumption in this study, the spatial distribution of AOD should be consistent with that of PM2.5, not PM10. 11. Page 13 line 22: Where is the J(NO2), surface, column? 12. Page 13 line 20 to Page 14 line 2: Other chemistry and physics processes may join with the photochemistry to impact tracer change in a nonlinear way.

References:

Bian, H., and C. S. Zender, Mineral dust and global tropospheric chemistry: The relative roles of photolysis and heterogeneous uptake. J. Geophys. Res., 108, 4672, 2003. Bian, H., and M. J. Prather, Accurate simulation of stratospheric photolysis in global chemical model. J. Atmos. Chem., 41, 281-296, 2002. Chin, M., T. Diehl, Q. Tian, J. M. Prospero, R. A. Kahn, A. Remer, H. Yu, A. M. Sayer, H. Bian, et al., Multi-decadal variations of atmospheric aerosols from 1980 to 2009: sources and regional trends, Atmos. Chem. Phys., 14, 3657-3690, doi:10.5194/acp-14-3657-2014, 2014. Wild, O., X. Zhu, and M. Prather, Fast-J: Accurate simulation of in- and below-cloud photolysis in tropospheric chemical models, J. Atmos. Chem. 37, 245-282, 2000.

---

## Referee Comment (RC2) · Anonymous Referee #2 · 12 Feb 2018

This study reports on impacts of solar brightening on summer-time ozone levels across Europe through analysis of model simulations examining the impact of changes in radiation on photolysis rates and biogenic emissions. Several studies have previously examined the impact of aerosol induced radiation perturbations on photolysis and subsequent atmospheric chemistry (Dickerson et al., 1997; Benas et al., 2007; Bian et al., 2007; Anger et al., 2016; Wang et al., 2016; Xing et al., 2017) and similar to the current study suggest that the aerosol induced reduction in solar irradiance leads to lower photolysis rates and less O3 production. Such interactions and feedbacks are a potentially important consideration for design of multi-pollutant control strategies seeking to simultaneously reduce O3 and particulate matter pollution. Thus studies that help quantify

the magnitude of these impacts relative to actual changes in composition of the atmosphere are of interest. Though the results reported are along expected lines, I think the manuscript needs to be strengthened to provide the context in which the results should be interpreted. In my assessment the current manuscript will benefit from some additional work in: (i) a clearer description of the design and methodology employed in the sensitivity experiments; (ii) clearer articulation of the assumptions and limitations of these experiments; and (iii) acknowledging that the study does not comprehensively examine the process changes induced by solar brightening between 1990 and 2010, but rather presents model sensitivity analyses that approximate the impact of aerosol burden changes on photolysis rates and biogenic emissions. The following comments and suggestions are offered:

1) The suggestion that the study examines changes in ozone between 1990 and 2010 due to solar brightening is misleading. Multiple atmospheric processes can be impacted by the direct radiative effects associated with brightening in addition to changes in photolysis rates (e.g., thermal reactions, atmospheric ventilation, changes in dry deposition). No specific simulations were conducted to fully represent conditions in 1990. Instead, AOD and surface shortwave radiation conditions "representative" of 1990's, approximated from changes in measured surface PM at a few sites, were used to perturb one component of the aerosol-radiation system, i.e., photolysis rates. Further, since no comparison of SSR changes over the time period are presented, it is difficult to ascertain whether the induced changes are actually representative of the brightening observed during this period. Thus, I would be careful in characterizing these results as trends or changes over the two decades. The analysis is essentially a sensitivity study and should be portrayed that way, so that the results can be conveyed and interpreted in an accurate manner. In addition to changes in the text, the authors could also consider an alternate, more representative title for the manuscript.

2) The model set-up and sensitivity simulation specifics could benefit from additional clarification. From what I understand, simulations with the CAMx model driven by meteorological fields from WRF and emissions representative of 2010 were first conducted (BASE run). Then a series of photolysis sensitivity simulations were conducted in which AOD used in the TUV photolysis code was somehow perturbed – this description currently is confusing and contradictory across the text.

a) Line 20-24 on page 5 first suggests that the TUV is used "externally" to estimate clear sky photolysis which are then adjusted in the model for clouds. It is also suggested that dry extinction efficiencies and SSA at 350nm are provided to the model. How is the AOD calculation then used to modify the already estimated clear sky photolysis rates? Line 26 on page 7 then suggests that the study used an in-line version of TUV? Which one is it? It seems that an in-line version of the TUV code in CAMx would be needed conduct the PHOT sensitivities described in Table 1, but from the current description it is not clear. Since much of the analysis focuses on these sensitivities, it is important that the model setup and the experimental design be clearly described.

b) Equation 2 shows how the AOD is estimated and modified. Some parts of the text suggest that sensitivities are approximating the impact of changes in aerosol burden on radiation and photolysis, which may lead readers to assume that the aerosol concentrations in the equation are being modified. However, I believe in the PHOT experiments the AOD is solely perturbed by the adjustment factor (pf). Please clarify.

c) If the perturbation is only induced through the adjustment factor, then the photolysis changes are only estimating the impacts on a chemical regime representative of 2010. I would imagine if (higher) emissions representative of the 1990s were used then the estimated changes in ozone due to the corresponding changes in photolysis would have been even larger.

d) Were photolysis rates through the model column perturbed by the same amount? Were the perturbations at the surface (or within the boundary layer) different from those in the free troposphere?

3) The authors should better explain the criteria for the choice of the observation locations used in estimating the trends in PM (section 3). Why were sites only in Switzerland and Netherlands (3 each) used and how they can be considered to be representative of regional PM trends across Europe?

4) Lines, 15-20 on Page 10 discuss the estimated trends in PM at the observation sites and quantify the changes to be 41-44%. How are these changes then used to estimate the 50% and 65% perturbations to the AOD for the sensitivity tests described on page 8?

5) Evaluation statistics for the BASE calculation are provided in Table 4, without much information on the measurements themselves -location, time, etc. Without such information it is difficult to gauge what these statistics represent. I believe correlation coefficient shown here is representative of the spatial variability captured by the model and not the "temporal evolution" as suggested on Page 19, line 30.

6) Section 4.3.2 and Figure 5: Please provide more details on how the SSR values are estimated. Are they from the WRF simulation or TUV? How different are the SSR from the two? Please emphasize and clarify the assumption that the changes in radiation only impact the photolysis rates and no other aspect of the modeled chemistry and transport.

7) Page 14, lines 1-2: I think the authors should caveat the conclusion that feedback chains associated with secondary aerosol formation and subsequent aerosol burden have negligible impact on photolysis rates. Direct radiative effects on temperature and boundary layer ventilation are also important effects that can modulate secondary organic aerosol production – since these effects are not accounted for in this study, I would caution against a broad conclusion.

8) The impacts of photolysis changes on seasonal average ozone mixing ratios are estimated to be rather modest (a few percent). I would imagine the impacts on daily maximum ozone values will be larger and would be of greater interest. It appears the authors have analyzed those impacts also, but have not presented them here. I think

many readers would be interested in impacts on daily maximum ozone.

9) The discussion in section 4.6 involving conversion of the change in ozone from the sensitivity runs to a trend over two decades and comparison with other reported trends is not convincing, especially given the range in the trends (0.06-0.16 ppb/yr). Given that the current study only examines the change induced by a single DRE process (i.e., photolysis) on a chemical state representative of 2010, I do not see how it can be converted and compared to a trend inferred from observations that have been influenced by many more chemical and physical processes that are not even approximated in this analysis.

10) A recent study by Xing et al. (2017) analyzes the impacts of aerosol direct effects on tropospheric ozone through changes in atmospheric dynamics and photolysis rates. For summertime conditions in China they report comparatively larger impacts on ambient ozone induced by DRE impacts on atmospheric dynamics (through stabilizing of the atmosphere and modulation of dry deposition) than photolysis. Their results suggest that reducing the aerosol DRE (as would happen in a brightening scenario) will benefit the reduction of maximum O3 in summer driven both by changes in photolysis and to a larger extent atmospheric dynamics. Could similar impacts of DRE changes also have occurred over Europe during the 1990-2010 brightening period?

References: Anger, A., Dessens, O., Xi, F., Barker, T., and Wu, R.: China's air pollution reduction efforts may result in an increase in surface ozone levels in highly polluted areas. Ambio, 45:254–265, 2016.

Benas, N., Mourtzanou, E., Kouvarakis, G., Bais, A., Mihalopoulos, N., and Vardavas, I.: Surface ozone photolysis rate trends in the Eastern Mediterranean: Modeling the effects of aerosols and total column ozone based on Terra MODIS data. Atmospheric Environment, 74, 1-9, 2013.

Bian, H., Han, S., Tie, X., Sun, M. and Liu, A.: Evidence of impact of aerosols on surface ozone concentration in Tianjin, China. Atmospheric Environment, 41(22), 4672-

4681, 2007

Dickerson, R.R., Kondragunta, S., Stenchikov, G., Civerolo, K.L., Doddridge, B.G. and Holben, B.N.: The impact of aerosols on solar ultraviolet radiation and photochemical smog. Science, 278(5339), 827-830, 1997.

Wang, J., Allen, D. J., Pickering, K. E., Li, Z. and He, H.: Impact of aerosol direct effect on East Asian air quality during the EAST-AIRE campaign, J. Geophys. Res. Atmos., 121, 2016, doi:10.1002/2016JD025108.

Xing, J., Wang, J., Mathur, R., Wang, S., Sarwar, G., Pleim, J., Hogrefe, C., Zhang, Y., Jiang, J., Wong, D. C., and Hao, J.: Impacts of aerosol direct effects on tropospheric ozone through changes in atmospheric dynamics and photolysis rates, Atmos. Chem. Phys., 17, 9869-9883, https://doi.org/10.5194/acp-17-9869-2017, 2017

---

## Author Comment (AC1) · 11 Apr 2018

**Responses to the comments of anonymous referee #1**

We would like to thank for the comments which helped to improve our manuscript. Please find below your comments in blue, our responses in black and modifications in the revised manuscript related to technical or specific comments in italic and inside quotes. In addition, we have updated the terminology of direct and indirect aerosol effects throughout the manuscript as well as in our replies to your comments, by replacing the term "direct aerosol radiative effect (DRE)" with "aerosol–radiation interactions (ARI)" and the term "aerosol indirect effects (AIE)" with "aerosol–cloud interactions (ACI)". All modifications are highlighted in the revised manuscript.

The authors explore the summer surface O3 change over Europe in response to the potential aerosol emission change from 1990 to 2010 via their impact on direct radiation effect (DRE) using CAMx air quality model. They expand their study by taking into account the feedback of biogenic emission through the chain of aerosol emission change to radiation change and perform all studies at different chemistry backgrounds (i.e. base and high NOx emissions). This is an interesting study and the study content is suitable for ACP. I recommend publishing the paper after the authors make minor modifications suggested below.

General Remarks: It would be good if the authors could explain why it is better to use the designed emissions in 1990 instead of using the emissions in 1990 provided directly from emission inventories. The authors use emission inventory TNO-MACC-III for 2010. To derive the emission in 1990, they first analyze the observed PM10 between 90s and 2010 over 3 Netherlands stations and 3 Switzerland stations to infer potential enhancement factors of emission in 1990 verses in 2010. They then generate three potential emissions representing the situation in 1990 by increasing the emissions in 2010 with these enhancement factors. However, there are emission inventories, such as A2-ACCMIP that can provide emissions directly back to 1980. A2-ACCMIP is one of the multi-year emission datasets available from the international initiative AeroCom project for its second phase (A2) hind-cast model experiments (http://aerocom.met.no) (Chin et al., 2014). It might also be good to claim the study as a sensitivity study of O3 change in response to aerosol emission change via DRE. The impact of solar "brightening" on surface ozone is not limited to aerosol DRE. Aerosol indirect effect (AIE) is another potential pathway as the authors discussed in the paper. Although the authors indicated that AIE is not a driving reason for surface solar radiation (SSR) trend over Europe during the study period based on some previous studies, AIE is still a non-negligible factor as suggested by a recent study (Mian Chin 2018, personal communication). Furthermore, aerosol can impact ozone chemistry via heterogeneous reaction and the effects of photolysis and heterogeneous update are typically nonlinear (Bian et al., 2003). Feedbacks of O3 production and loss due to changes in O3 precursors leads to the complicated nonlinear feature.

We think that there was a misunderstanding about our approach to test the impact on ozone via the photolysis rates. The PHOT1−3 scenarios have the same emissions and meteorological input as the base case (BASE scenario). Therefore, we did not adjust any PM emissions in the three reported sensitivity tests, PHOT1−3. What we did was to adjust the PM only in the AOD calculations inside CAMx (in the in-line TUV), as described in Sect. 2.3.1. In other words, the induced forcing in the PHOT1−3 scenarios lies only in the AOD calculations. This approach excludes any potential impacts on the gas-aerosol chemistry, which would be the case if we would have adjusted the PM emissions or the PM concentrations generally in CAMx (and not just in the AOD calculation as we did). For example, consider the eq. (2) from the manuscript. Let $C$ be the PM concentration species for the base case (BASE scenario) and $p_f$ the adjustment factor that is applied to each PM species concentration. The product of $C \cdot p_f$ yields a new adjusted PM species concentration (that exists only in the AOD calculation) which we denote here as (referring to any of the PHOT1−3 scenarios) $C_{phot} = C \cdot p_f$. So, the actual concentration of PM species, $C$, does not change, except for secondary aerosols due to indirect impact of changes in the chemistry originating from the changes in the photolysis rates. However, we investigated this exception and we showed that the change in secondary aerosol is small (see Figs. S3 and S4) and any subsequent change in the AOD and the photolysis rates is expected to be negligible.

Although it was not our intention to use 1990 emissions in this study, we would like to point out that there are no reliable PM emissions in Europe before 2000 to use in regional models and those emissions that were estimated using various assumptions and gap-filling procedures for years before 2000 (for trend analyses) have very high uncertainties (Colette et al., 2016).

Specific comments 1. Page 1 lines 13-14 in Abstract: PHOT1, PHOT2, and PHO#3 is not in the effects of increased radiation on photolysis rates.

For a better clarification we reformulated the part of the sentence on page 1, lines 15−18 as follows:

*"… the effects of increased radiation between 1990 and 2010 on photolysis rates (with the PHOT1, PHOT2 and PHOT3 scenarios, which represented the radiation in 1990) and on biogenic volatile organic compounds (BVOCs) emissions (with the BIO scenario, which represented the biogenic emissions in 1990), …"*

2. Page 2 line 14: φ and σ depend not only on the gaseous species and air temperature, but also on air pressure for some VOCs.

We modified the sentence on page 2, lines 14-15 as follows:

*"φ and σ depend on the gaseous species and the air temperature T (K) as well as on air pressure for some species …"*

3. Page 2 line 15: Please add "O2, O3 and" before "water vapor". Please also add references of Bian et al., 2002 and Wild et al., 2000 at the end of this sentence.

Your suggestions were implemented in the manuscript (page 2, line 17).

4. Page 5 line 9: Is the two-weeks long enough spin-up for O3 chemistry?

Berge et al. (2001) showed that the impact of initial conditions on ozone is reduced to 10% or less with a spin-up time of approximately 3 days, while Jiménez et al. (2007) showed the same but for a spin-up time of 48 h. Other air quality modeling studies (Streets et al., 2007; Godowitch et al., 2008; Czader et al., 2012) focusing on ozone chemistry have used spin-up times between 2 and 10 days for simulating the summer season or summer episodes. Therefore, we are confident that the 2 weeks spin-up time was long enough.

5. Page 7 line 14: Could you elaborate on how to increase NOx emission?

As also stated in Table 1 in the manuscript, the $NO_x$ emissions were increased by a factor of 2 for each SNAP (Selected Nomenclature for Air Pollution) category based on the study of Oikonomakis et al. (2018). We added this clarification in Table 1.

6. Page 7 line 22: Do you really use the same emissions for both cases?

As also discussed above we used the same anthropogenic emissions and meteorology for all cases. For the PHOT1−3 scenarios we also used the same biogenic emissions, while for the BIO scenario the biogenic emissions were modified. We modified the sentence on page 8, lines 1−3 as follows:

*"In other words, we used the same meteorology and emissions for both cases (except for the BVOC emission sensitivity tests where we used different BVOC emissions) ..."*

7. Page 9 line 29: It seems to me that you are not holding the aerosol coarse fraction constant during the period, but holding the absolute coarse model aerosol amount unchanged.

By "aerosol coarse fraction" we mean the particles with an aerodynamic diameter $2.5 < d \leq 10$. In CAMx, for the simulation of PM only two modes are considered: a fine one ($d \leq 2.5$) and a coarse one ($2.5 < d \leq 10$). The latter one we did not

adjust/change in the PHOT1−3 scenarios and therefore it remained constant in line with our assumptions in page 9, line 29. In order to make our statements and assumptions more clear, we replaced the "aerosol coarse fraction" with "aerosol coarse mode", wherever it was mentioned in the manuscript.

8. Page 10 lines 4-6: The criteria III should not be listed here since all PM2.5 data in this study are estimated based on PM10, while not measured.

The criteria iii is about the PM10 and PM2.5 data availability for 2010. In Sect. 2.2 we mentioned that both the $PM_{10}$ and $PM_{2.5}$ were provided by Airbase and NABEL networks and both products were measured at the respective sites.

9. Page 11 line 28: What aerosol components are included in FPRM?

FPRM (fine other primary) in CAMx refers to fine (<2.5 μm in diameter) primary particles other than those explicitly represented in the emissions and in the CAMx model. In the TNO-MAC-III emission inventory used in this study, the PM emissions are split to the following components: EC, OC, $Na^+$, $SO_4^{2-}$ and "Other minerals". After the fractions of the first four species are calculated, the remaining part is assigned to the "Other minerals", which contains other non-carbonaceous particles (Kuenen et al., 2014) and named as FPRM in CAMx.

10. Page 12 lines 18-20: To support the assumption in this study, the spatial distribution of AOD should be consistent with that of PM2.5, not PM10.

Ideally, we would also like to compare the spatial distribution of our simulated AOD changes with the respective ones of $PM_{2.5}$ concentrations. However, in Colette et al. (2017) only the simulated spatial distribution of the $PM_{10}$ concentration changes was shown. On the other hand, as discussed in Sect. 3, it was suggested that the decreasing trend in $PM_{10}$ concentrations was dominated by the reductions in the $PM_{2.5}$ concentrations (Barmpadimos et al., 2012; Tørseth et al., 2012), and thus we expect that the spatial distribution of the changes in $PM_{10}$ concentration to be similar to that of $PM_{2.5}$ concentrations. Moreover, Tørseth et al. (2012) showed that the geographical distribution between the changes in $PM_{10}$ and $PM_{2.5}$ concentrations, for the period 2000−2009, was similar for several EMEP monitoring sites. Therefore, we believe that the spatial distribution of the changes in $PM_{10}$ concentrations is a good proxy for that in $PM_{2.5}$ concentrations, especially for the areas which are not strongly influenced by sea salt and African dust particles.

11. Page 13 line 22: Where is the J(NO2), surface, column?

The $J(NO_2)$, is reported at ground-level. We made the following modifications in the text and figure captions:

page 14, line 4: *"The simulated (in the base case) ground-level photolysis rate of NO₂, J(NO₂), ..."*

page 14, lines 5−6: *"The modeled continental mean absolute (relative) differences in ground-level J(NO₂) ..."*

page 14, lines 7−8: *"The spatial distribution and relative changes are the same for the ground-level photolysis rate of O₃, J(O₃ → O¹D), ..."*

caption of Figure 6: *"Seasonal daytime (10:00−18:00 LMT) mean J(NO₂) at ground level ..."*

caption of Figure S3 in the Supplement: *"Seasonal daytime (10:00-18:00 LMT) mean J(O₃ → O¹D) at ground level ..."*

12. Page 13 line 20 to Page 14 line 2: Other chemistry and physics processes may join with the photochemistry to impact tracer change in a nonlinear way.

That is true, but in this study with the PHOT1−3 scenarios we investigated and isolated only the impact of aerosol–radiation interactions (ARI) on the chemistry and only via photolysis rates, excluding the impact of ARI on meteorological processes.

**References**

Barmpadimos, I., Keller, J., Oderbolz, D., Hueglin, C., and Prévôt, A. S. H.: One decade of parallel fine (PM$_{2.5}$) and coarse (PM$_{10}$–PM$_{2.5}$) particulate matter measurements in Europe: trends and variability, Atmos. Chem. Phys., 12, 3189-3203, doi:10.5194/acp-12-3189-2012, 2012.

Berge, E., Huang, H.-C., Chang, J., and Liu, T.-H.: A study of the importance of initial conditions for photochemical oxidant modeling, J. Geophys. Res, 106, 1347-1363, doi:10.1029/2000JD900227, 2001.

Colette, A., Aas, W., Banin, L., Braban, C. F., Ferm, M., González Ortiz, A., Ilyin, I., Mar, K., Pandolfi, M., Putaud, J.-P., Shatalov, V., Solberg, S., Spindler, G., Tarasova, O., Vana, M., Adani, M., Almodovar, P., Berton, E., Bessagnet, B., Bohlin-Nizzetto, P., J., B., Breivik, K., Briganti, G., Cappelletti, A., Cuvelier, K., Derwent, R., D'Isidoro, M., Fagerli, H., Funk, C., Garcia Vivanco, M., Haeuber, R., Hueglin, C., Jenkins, S., Kerr, J., de Leeuw, F., Lynch, J., Manders, A., Mircea, M., Pay, M. T., Pritula, D., Querol, X., Raffort, V., Reiss, I., Roustan, Y., Sauvage, S., Scavo, K., Simpson, D., Smith, R. I., Tang, Y. S., Theobald, M., Tørseth, K., Tsyro, S., van Pul, A., Vidic, S., Wallasch, M., and Wind, P.: Air Pollution Trends in the EMEP Region between 1990 and 2012, EMEP/CCC-Report 1/2016, 2016.

Colette, A., Andersson, C., Manders, A., Mar, K., Mircea, M., Pay, M. T., Raffort, V., Tsyro, S., Cuvelier, C., Adani, M., Bessagnet, B., Bergström, R., Briganti, G., Butler, T., Cappelletti, A., Couvidat, F., D'Isidoro, M., Doumbia, T., Fagerli, H., Granier, C., Heyes, C., Klimont, Z., Ojha, N., Otero, N., Schaap, M., Sindelarova, K., Stegehuis, A. I., Roustan, Y., Vautard, R., van Meijgaard, E., Vivanco, M. G., and Wind, P.: EURODELTA-Trends, a multi-model experiment of air quality hindcast in Europe over 1990–2010, Geosci. Model Dev., 10, 3255-3276, doi:10.5194/gmd-10-3255-2017, 2017.

Czader, B. H., Rappenglück, B., Percell, P., Byun, D. W., Ngan, F., and Kim, S.: Modeling nitrous acid and its impact on ozone and hydroxyl radical during the Texas Air Quality Study 2006, Atmos. Chem. Phys., 12, 6939-6951, doi:10.5194/acp-12-6939-2012, 2012.

Godowitch, J. M., Gilliland, A. B., Draxler, R. R., and Rao, S. T.: Modeling assessment of point source NOx emission reductions on ozone air quality in the eastern United States, Atmos. Environ., 42, 87-100, https://doi.org/10.1016/j.atmosenv.2007.09.032, 2008.

Jiménez, P., Parra, R., and Baldasano, J. M.: Influence of initial and boundary conditions for ozone modeling in very complex terrains: A case study in the northeastern Iberian Peninsula, Environmental Modelling & Software, 22, 1294-1306, https://doi.org/10.1016/j.envsoft.2006.08.004, 2007.

Kuenen, J., Visschedijk, A., Jozwicka, M., and Denier Van Der Gon, H.: TNO-MACC_II emission inventory; a multi-year (2003–2009) consistent high-resolution European emission inventory for air quality modelling, Atmos. Chem. Phys., 14, 10963-10976, doi:10.5194/acp-14-10963-2014, 2014.

Oikonomakis, E., Aksoyoglu, S., Ciarelli, G., Baltensperger, U., and Prévôt, A. S. H.: Low modeled ozone production suggests underestimation of precursor emissions (especially NOx) in Europe, Atmos. Chem. Phys., 18, 2175-2198, doi:10.5194/acp-18-2175-2018, 2018.

Streets, D. G., Fu, J. S., Jang, C. J., Hao, J., He, K., Tang, X., Zhang, Y., Wang, Z., Li, Z., Zhang, Q., Wang, L., Wang, B., and Yu, C.: Air quality during the 2008 Beijing Olympic Games, Atmos. Environ., 41, 480-492, https://doi.org/10.1016/j.atmosenv.2006.08.046, 2007.

Tørseth, K., Aas, W., Breivik, K., Fjæraa, A. M., Fiebig, M., Hjellbrekke, A. G., Lund Myhre, C., Solberg, S., and Yttri, K. E.: Introduction to the European Monitoring and Evaluation Programme (EMEP) and observed atmospheric composition change during 1972–2009, Atmos. Chem. Phys., 12, 5447-5481, doi:10.5194/acp-12-5447-2012, 2012.

---

## Author Comment (AC2) · 11 Apr 2018

**Responses to the comments of anonymous referee #2**

We would like to thank for the comments which helped to improve our manuscript. Please find below your comments in blue, our responses in black and modifications in the revised manuscript related to technical or specific comments in italic and inside quotes. In addition, we have updated the terminology of direct and indirect aerosol effects throughout the manuscript as well as in our replies to your comments, by replacing the term "direct aerosol radiative effect (DRE)" with "aerosol–radiation interactions (ARI)" and the term "aerosol indirect effects (AIE)" with "aerosol–cloud interactions (ACI)". All modifications are highlighted in the revised manuscript.

This study reports on impacts of solar brightening on summer-time ozone levels across Europe through analysis of model simulations examining the impact of changes in radiation on photolysis rates and biogenic emissions. Several studies have previously examined the impact of aerosol induced radiation perturbations on photolysis and subsequent atmospheric chemistry (Dickerson et al., 1997; Benas et al., 2007; Bian et al., 2007; Anger et al., 2016; Wang et al., 2016; Xing et al., 2017) and similar to the current study suggest that the aerosol induced reduction in solar irradiance leads to lower photolysis rates and less O3 production. Such interactions and feedbacks are a potentially important consideration for design of multi-pollutant control strategies seeking to simultaneously reduce O3 and particulate matter pollution. Thus studies that help quantify the magnitude of these impacts relative to actual changes in composition of the atmosphere are of interest. Though the results reported are along expected lines, I think the manuscript needs to be strengthened to provide the context in which the results should be interpreted. In my assessment the current manuscript will benefit from some additional work in: (i) a clearer description of the design and methodology employed in the sensitivity experiments; (ii) clearer articulation of the assumptions and limitations of these experiments; and (iii) acknowledging that the study does not comprehensively examine the process changes induced by solar brightening between 1990 and 2010, but rather presents model sensitivity analyses that approximate the impact of aerosol burden changes on photolysis rates and biogenic emissions. The following comments and suggestions are offered:

1) The suggestion that the study examines changes in ozone between 1990 and 2010 due to solar brightening is misleading. Multiple atmospheric processes can be impacted by the direct radiative effects associated with brightening in addition to changes in photolysis rates (e.g., thermal reactions, atmospheric ventilation, changes in dry deposition). No specific simulations were conducted to fully represent conditions in 1990. Instead, AOD and surface shortwave radiation conditions "representative" of 1990's, approximated from changes in measured surface PM at a few sites, were used to perturb one component of the aerosol-radiation system, i.e., photolysis rates. Further, since no comparison of SSR changes over the time period are presented, it is difficult to ascertain whether the induced changes are actually representative of the brightening

observed during this period. Thus, I would be careful in characterizing these results as trends or changes over the two decades. The analysis is essentially a sensitivity study and should be portrayed that way, so that the results can be conveyed and interpreted in an accurate manner. In addition to changes in the text, the authors could also consider an alternate, more representative title for the manuscript.

Thank you for your suggestions. First, we would like to clarify that it was our intention not to use the emissions and meteorology of 1990 in order to isolate the impact of aerosol–radiation interactions (ARI) on photolysis rates and biogenic emissions, which cannot be the case if the emissions and meteorology of 1990 are used. Consequently, our goal was not to simulate the actual conditions of 1990, but the aerosol optical depth (AOD) and surface solar radiation (SSR) conditions representative of 1990. That is why we compared our simulated AOD and SSR changes with other modeling and observational studies, in Sections 4.3.1 and 4.3.2, to test the validity and representation of our modeling experiments. As discussed and shown in Sections 4.3.1 and 4.3.2, our simulated AOD and SSR changes are within the range of the respective changes reported by other modeling and observational studies for either the same time period or other time periods similar to the one examined in this study.

We made several changes in the text to present in a more clear way the setup and assumptions of our modeling experiments as well as the context in which these results should be interpreted. Some of these changes are listed in the replies of your next comments as well as in the replies of the 1st anonymous reviewer and some are listed below:

from page 1 line 32 to page 2 line 1: *"Finally, the role of the aerosol–radiation interactions (ARI) changes in the European summer surface ..."*

page 9 line 32: *"... to assess the overall impact of the ARI changes on surface ozone"*

page 16 line 11: *"We investigated the impact of the ARI changes on European summer surface ozone between 1990 and 2010 using the ..."*

page 17 line 6: *"Nevertheless, the role of the ARI changes (as quantified in this study) ..."*

We also took into consideration your suggestion of modifying the title of the manuscript into a different one, which reflects more accurately the work that has been done in this study:

*"Solar "brightening" impact on summer surface ozone between 1990 and 2010 in Europe – A model sensitivity study of the influence of the aerosol–radiation interactions"*

2) The model set-up and sensitivity simulation specifics could benefit from additional clarification. From what I understand, simulations with the CAMx model driven by meteorological fields from WRF and emissions representative of 2010 were

first conducted (BASE run). Then a series of photolysis sensitivity simulations were conducted in which AOD used in the TUV photolysis code was somehow perturbed – this description currently is confusing and contradictory across the text.

a) Line 20-24 on page 5 first suggests that the TUV is used "externally" to estimate clear sky photolysis which are then adjusted in the model for clouds. It is also suggested that dry extinction efficiencies and SSA at 350nm are provided to the model. How is the AOD calculation then used to modify the already estimated clear sky photolysis rates? Line 26 on page 7 then suggests that the study used an in-line version of TUV? Which one is it? It seems that an in-line version of the TUV code in CAMx would be needed conduct the PHOT sensitivities described in Table 1, but from the current description it is not clear. Since much of the analysis focuses on these sensitivities, it is important that the model setup and the experimental design be clearly described.

The CAMx model first requires the photolysis rates in clear-sky conditions, which are calculated by the TUV model and fed as input into CAMx. In the second step, the clear-sky photolysis rates are adjusted for clouds, aerosols, temperature and pressure in one of the CAMx subroutines called "in line TUV". We modified the sentence on page 5, lines 25−26 as follows:

*"Then, these rates are internally adjusted in CAMx every hour for clouds and aerosols (simulated by CAMx) …"*

The dry extinction efficiency and single scattering albedo (SSA) values at 350 nm are provided only in the in-line version of TUV, as no aerosol species are considered in the external TUV. We modified the text on page 6, lines 2−5 as follows:

*"Inside CAMx, the COD is calculated for each model grid cell based on the approach of Genio et al. (1996) and Voulgarakis et al. (2009), while the dry extinction efficiency of the aerosol species, which is needed for the calculation of the AOD, as well as the single-scattering albedo (SSA) were provided by Takemura et al. (2002) for the wavelength of 350 nm (Table S1)."*

The internal adjustment for clouds and aerosols in CAMx is performed into two steps: First, the clear-sky shortwave solar radiation and photolysis rates are re-calculated, but this time only for a single representative wavelength of 350 nm. Then, the radiative calculations are repeated including in this second step the impact of clouds and aerosols. A ratio of clear-to-cloudy (and aerosols) sky is derived by the aforementioned radiative calculations in CAMx and then applied to the calculation of clear-sky photolysis rates and shortwave solar radiation in the TUV model. The following modification was applied in the text on page 5, lines 27−32:

*"The internal adjustment for clouds and aerosols inside CAMx is performed into two steps: First, the clear-sky shortwave solar radiation and photolysis rates are re-calculated inside CAMx, but this time only for a single representative wavelength of 350 nm. In the second step, the radiative transfer calculations are repeated including the impact of clouds and aerosols. A ratio of clear-to-cloudy (and aerosols) sky is derived by the aforementioned radiative transfer calculations in CAMx, which is applied to the clear-sky photolysis rates and shortwave solar radiation that were calculated by TUV and were given as input to CAMx."*

We hope that these clarifications describe the induced AOD perturbations within CAMx more clearly.

b) Equation 2 shows how the AOD is estimated and modified. Some parts of the text suggest that sensitivities are approximating the impact of changes in aerosol burden on radiation and photolysis, which may lead readers to assume that the aerosol concentrations in the equation are being modified. However, I believe in the PHOT experiments the AOD is solely perturbed by the adjustment factor (pf). Please clarify.

For the PHOT1−3 scenarios we adjusted the PM concentrations only in the calculation of AOD inside the CAMx model, as described in Sect. 2.3.1. In other words, the induced forcing in the PHOT1−3 scenarios lies only in the AOD calculation. For example, consider the eq. (2) from the manuscript. Let $C$ be the PM concentration species for the base case (BASE scenario) and $p_f$ the adjustment factor that is applied to each PM species concentration. The product of $C \cdot p_f$ yields a new adjusted PM species concentration (which exists only in the AOD calculation) which we denote here (referring to any of the PHOT1−3 scenarios) as $C_{phot} = C \cdot p_f$. So, the actual PM concentration $C$ does not change, except for the case of secondary aerosols due to indirect impact of changes in the chemistry originating from the changes in the photolysis rates. However, we investigated this exception and we showed that the change in secondary aerosol is small (see Figs. S3 and S4) and any subsequent change in the AOD and the photolysis rates is expected to be negligible. Therefore, the adjustment factor $p_f$ gives a pseudo-perturbation of changes in PM concentrations, since the PM concentrations do not actually change but at the same time we are able to isolate the radiative impact of PM concentration changes (as AOD is perturbed) on solar radiation and photolysis rates. This is why in some parts of the text it is suggested that the PHOT1−3 scenarios are approximating the impact of changes in aerosol burden on solar radiation and photolysis rates.

We modified the sentence on page 8, lines 9−10 as follows:

*"In order to quantify only the changes in ARI, we isolated them from other aerosol effects such as the gas-aerosol chemical interactions"*

We added a sentence on page 8, lines 15−17:

*"Hence, the product $p_f \cdot C$ represents the PM concentrations in 1990, but purely in AOD calculations in order to generate only AOD, solar radiation and photolysis rates in 1990."*

c) If the perturbation is only induced through the adjustment factor, then the photolysis changes are only estimating the impacts on a chemical regime representative of 2010. I would imagine if (higher) emissions representative of the 1990s were used then the estimated changes in ozone due to the corresponding changes in photolysis would have been even larger.

In Sect. 2.3 we wrote that we performed all the sensitivity runs also using a second base case with doubled $NO_x$ emissions (BASE_$NO_x$ scenario) compared to the initial base case (BASE scenario). The original motivation for this addition was a potential underestimation of ozone precursor emissions in the inventories, as discussed in Oikonomakis et al. (2018). However, the results of the scenarios using the base case with increased $NO_x$, can also give an indication of the impact of changes in ARI on photolysis rates and ozone in a chemical regime different than that of 2010. Our results (Fig. 8) indicate that using a different chemical regime leads mainly to a larger spatial coverage of the impact (of changes in ARI) on ozone, while the magnitude of those effects is influenced to a lesser degree.

d) Were photolysis rates through the model column perturbed by the same amount? Were the perturbations at the surface (or within the boundary layer) different from those in the free troposphere?

As shown in eq. (2), the AOD and hence the photolysis rates were perturbed by the relative same amount through the whole model column. However, since the PM concentrations decrease rapidly with altitude, the induced perturbation in the PM concentrations in the calculation of AOD will be higher in absolute terms closer to the surface and very small higher up in the free troposphere. In other words, since the applied perturbations in eq. (2) are in relative terms, this implies that the magnitude of these perturbations in absolute terms follows the magnitude of the PM concentrations.

3) The authors should better explain the criteria for the choice of the observation locations used in estimating the trends in PM (section 3). Why were sites only in Switzerland and Netherlands (3 each) used and how they can be considered to be representative of regional PM trends across Europe?

The first and most important selection criterion for the sites was to have available $PM_{10}$ measurements (or equivalent products) as early as possible, close to the year 1990. This was only the case for Switzerland with corrected total suspended particle (TSP) as proxies for $PM_{10}$ concentrations. In addition, we decided to include the Netherlands as well, since the data in the monitoring sites were available since 1992. Other monitoring sites in Europe did not have PM data going back to beginning of the 1990s, as reported by Airbase.

Tørseth et al. (2012) and Barmpadimos et al. (2012) have shown that the decreasing trends in both $PM_{10}$ and $PM_{2.5}$ concentrations in Switzerland were similar to other western and central European countries for the time periods 2000–2009 and 1998–2010, respectively. Furthermore, Hoogerbrugge et al. (2010) showed that the decreasing trends in $PM_{10}$ concentrations for the Netherlands were also similar to other western and central European countries (i.e. Belgium, France, Germany, United Kingdom and Denmark) for the period 1990–2006. Therefore, we believe that the results of PM trend analyses for Switzerland and the Netherlands in this study provided a good representation of the PM changes in most parts of

Europe. On the other hand, the PM changes reported in Table 2 might not be representative for southeastern Europe and the results of the sensitivity analysis might be an overestimation of the true changes, as also stated in Sect. 5.

4) Lines, 15-20 on Page 10 discuss the estimated trends in PM at the observation sites and quantify the changes to be 41-44%. How are these changes then used to estimate the 50% and 65% perturbations to the AOD for the sensitivity tests described on page 8?

The numbers 41-43% refer to annual changes in PM10. The calculated perturbations in the AOD were based on the values for PM2.5 in summer in Table 2. More specifically, 53% (for the perturbation we rounded it down to 50%) and 65% relative changes of the estimated $PM_{2.5}$ concentrations (reported in parentheses) in the summer season for Switzerland and the Netherlands, respectively. We modified the sentence on page 10, lines 15−16 as follows:

*"The adjustment factors ($p_f$) were then based on the total relative changes of the estimated $PM_{2.5}$ concentrations for the summer season (see Table 2)."*

5) Evaluation statistics for the BASE calculation are provided in Table 4, without much information on the measurements themselves -location, time, etc. Without such information it is difficult to gauge what these statistics represent. I believe correlation coefficient shown here is representative of the spatial variability captured by the model and not the "temporal evolution" as suggested on Page 10, line 30.

We revised the statistical model evaluation for $O_3$, $PM_{10}$ and $PM_{2.5}$ by reducing the data availability filter from 90% (which was used in Oikonomakis et al. (2018)) to 80% (which had already been used for AOD and SSR) in order to include some more stations, and hence increase the spatial representation of the model evaluation, but without degrading its performance. The changes are small and only in a few statistical metrics, and are highlighted in Table 4 of the revised manuscript. In addition, we added a figure in the supplement (Fig. S1) which shows the correlation coefficient for each station and for each evaluated variable in a map. The correlation coefficient for most stations in Fig. S1 is equal or higher than the overall correlation coefficient reported in Table 4, indicating that the model successfully catches most of the temporal evolution of all variables and not just the spatial variability. We made the following changes in the manuscript regarding Fig. S1 and also to provide some more information about the treatment of the measurements:

page 7, lines 6−8: *"For a better comparison between the model and the observations, we used only rural background stations due to our grid resolution. Furthermore, we evaluated the daily mean of the chemical species in order to be able to compare our results with other studies (e.g. Bessagnet et al., 2016). More details about the observational data treatment and ..."*

page 11, line 8: *"... for the daily mean $O_3$, $PM_{2.5}$ and $PM_{10}$ (see also Fig. S1)."*

page 11, line 11: *"... the model can capture the observed $PM_{10}$ temporal evolution (Fig. S1)."*

page 11, line 20: *"... relatively high correlation (r = 0.6) between the model and the observations which is shown in more detail in Fig. S1."*

page 11, line 28: *"... inter-daily variability was captured as well (Figures 2 and S1; r = 0.8)."*

6) Section 4.3.2 and Figure 5: Please provide more details on how the SSR values are estimated. Are they from the WRF simulation or TUV? How different are the SSR from the two? Please emphasize and clarify the assumption that the changes in radiation only impact the photolysis rates and no other aspect of the modeled chemistry and transport.

In Sect. 2.1 (page 6, line 14) it is stated that the algorithms in both TUV and CAMx were modified to just extract the AOD and SSR data. Therefore, the SSR results presented in this study come from the modeling system of TUV-CAMx and were calculated with the same radiative transfer algorithm that was used for the calculation of photolysis rates (as described in detail in Sect. 2.1). The SSR originating from WRF is only used in the calculation of biogenic emissions. A consistent comparison of SSR between WRF and TUV-CAMx is not possible, as the first has instantaneous hourly output and the latter has hourly average output. However, in both cases the same cloud cover data are used in the radiative transfer calculations, and thus we do not expect significant differences in the SSR between the two models. We made the following modifications in the text to make this more clear:

page 6, lines 15−17: *"In addition, the radiative transfer algorithms of both TUV and CAMx were modified to extract the modeled AOD and SSR data. In other words, both the SSR (used in the photolysis rate calculation) and the photolysis rates were calculated according to the same parameterization that was described above."*

page 6, lines 31−32: *"... using temperature and SSR data from the WRF output (the SSR data from WRF were not used in any calculation in CAMx) as well as land use data from ..."*

page 8, lines 4−7: *"Finally, it is noted that the chemistry simulated by CAMx (for any scenario) does not affect the meteorology, as it is prescribed (see Sect. 2.1), and hence the impact of ARI on atmospheric dynamics and other meteorological related effects (e.g. vertical mixing, dry deposition (Xing et al., 2017)) are excluded in this study."*

7) Page 14, lines 1-2: I think the authors should caveat the conclusion that feedback chains associated with secondary aerosol formation and subsequent aerosol burden have negligible impact on photolysis rates. Direct radiative effects on temperature

and boundary layer ventilation are also important effects that can modulate secondary organic aerosol production – since these effects are not accounted for in this study, I would caution against a broad conclusion.

We have modified that sentence to explicitly state that the presented argument about the secondary aerosol changes is relevant only to the results of this study (or similar studies within the same framework), which does not take into account any potential effects of ARI on meteorology and subsequent effects on aerosol chemistry.

page 14, lines 14−16: *"We conclude that these changes in SA have negligible impact on the photolysis rates. However, the changes in SA might not be negligible if the impact of ARI on meteorology and subsequent effects on chemistry are also taken into account (which is not the case for this study)."*

8) The impacts of photolysis changes on seasonal average ozone mixing ratios are estimated to be rather modest (a few percent). I would imagine the impacts on daily maximum ozone values will be larger and would be of greater interest. It appears the authors have analyzed those impacts also, but have not presented them here. I think many readers would be interested in impacts on daily maximum ozone.

The impact of ARI changes on daily maximum ozone is higher only by up to 0.1 ppb, as shown in Fig. 1 below, compared to the daytime (10:00−18:00 LMT) average results that we reported in the manuscript. The reason is that ozone peaks in the late afternoon hours, while the maximum impact of the ARI on ozone occurs earlier. Therefore, we think that is not necessary to include the results of Fig. 1 in the manuscript, as they do not bring any additional interesting information to our study.

[Figure]

**Figure 1.** Seasonal mean of daily maximum O$_3$ mixing ratios for the BASE scenario (a) and differences in daily maximum O$_3$ between the BASE scenario and PHOT1, PHOT2 and PHOT3 scenarios (b-d), respectively, in summer 2010. Note the different color scale between panel (a) and panels (b-d).

9) The discussion in section 4.6 involving conversion of the change in ozone from the sensitivity runs to a trend over two decades and comparison with other reported trends is not convincing, especially given the range in the trends (0.06-0.16 ppb/yr). Given that the current study only examines the change induced by a single DRE process (i.e., photolysis) on a chemical state representative of 2010, I do not see how it can be converted and compared to a trend inferred from observations that have been influenced by many more chemical and physical processes that are not even approximated in this analysis.

As we explicitly stated in Sect. 4.6, we did not compare the changes in ozone reported in this study with the results of observed ozone trends with other studies. We just wanted to show a very general comparison between the orders of magnitude between our results, the total ozone concentrations and the observed ozone trends. We made the following modifications to Sect. 4.6:

page 15, line 27, section's title: *"ARI and ozone trends"*

from page 15, line 30 to page 16 line 6: *"Wilson et al. (2012) reported an annual (summer) increasing trend of 0.16 ± 0.02 (0.12 ± 0.06) ppb yr$^{-1}$ in the European ground-level ozone (stations-average) for the period 1996-2005. The total ozone difference (0.2–0.8 ppb) via both the effects on photolysis rates and BVOCs emissions (COMBO scenario) would translate (considering the full 20-year time period) to a summer trend of 0.01–0.04 ppb yr$^{-1}$. These values should not be considered for a direct comparison with the absolute values of the aforementioned observed ozone trends, not only due to differences in the data analysis like time averaging and spatial coverage, but most importantly due to the exclusion of other physical and chemical processes influencing the ozone trends. Nevertheless, the comparison of the order of magnitude between the aforementioned values and the reported ozone trends suggests a higher importance of the impact of ARI (only via photolysis rates and BVOCs emissions) on surface ozone than when just comparing to the total ozone concentrations. Therefore, this comparison indicates that the ARI (as investigated in this study) might have had an accountable impact on …"*

10) A recent study by Xing et al. (2017) analyzes the impacts of aerosol direct effects on tropospheric ozone through changes in atmospheric dynamics and photolysis rates. For summertime conditions in China they report comparatively larger impacts on ambient ozone induced by DRE impacts on atmospheric dynamics (through stabilizing of the atmosphere and modulation of dry deposition) than photolysis. Their results suggest that reducing the aerosol DRE (as would happen in a brightening scenario) will benefit the reduction of maximum O3 in summer driven both by changes in photolysis and to a larger extent atmospheric dynamics. Could similar impacts of DRE changes also have occurred over Europe during the 1990-2010 brightening period?

We have already discussed the results of other studies that included the ARI on weather or ACI and their subsequent effects on chemistry in Sect. 1 as well as in the conclusions (Sect. 5), in detail. It could be that the reduction of the ARI, via the effects on atmospheric dynamics, might have resulted in reduction in summer maximum surface ozone over Europe between 1990 and 2010, as shown by Xing et al. (2015). However, there are still a lot of uncertainties in the online models and especially in the PBL mixing processes (Baklanov et al., 2014), which were discussed in detail in Sect. 1 of the manuscript. In addition, we believe that further investigation on the topic with different models and parameterizations, as well as improvement of the models with observational and experimental information, is needed to increase our confidence in the model results in terms of agreement with the reality.

**References**

Baklanov, A., Schlünzen, K., Suppan, P., Baldasano, J., Brunner, D., Aksoyoglu, S., Carmichael, G., Douros, J., Flemming, J., Forkel, R., Galmarini, S., Gauss, M., Grell, G., Hirtl, M., Joffre, S., Jorba, O., Kaas, E., Kaasik, M., Kallos, G., Kong, X., Korsholm, U., Kurganskiy, A., Kushta, J., Lohmann, U., Mahura, A., Manders-Groot, A., Maurizi, A., Moussiopoulos, N., Rao, S. T., Savage, N., Seigneur, C., Sokhi, R. S., Solazzo, E., Solomos, S., Sørensen, B., Tsegas, G., Vignati, E., Vogel, B., and Zhang, Y.: Online coupled regional meteorology chemistry models in Europe: current status and prospects, Atmos. Chem. Phys., 14, 317-398, doi:10.5194/acp-14-317-2014, 2014.

Barmpadimos, I., Keller, J., Oderbolz, D., Hueglin, C., and Prévôt, A. S. H.: One decade of parallel fine ($PM_{2.5}$) and coarse ($PM_{10}$–$PM_{2.5}$) particulate matter measurements in Europe: trends and variability, Atmos. Chem. Phys., 12, 3189-3203, doi:10.5194/acp-12-3189-2012, 2012.

Hoogerbrugge, R., van der Gon, H. D., van Zanten, M., and Matthijsen, J.: Trends in Particulate Matter. Report 500099014/2010, PBL Netherlands Environmental Assessment Agency, 2010.

Oikonomakis, E., Aksoyoglu, S., Ciarelli, G., Baltensperger, U., and Prévôt, A. S. H.: Low modeled ozone production suggests underestimation of precursor emissions (especially NOx) in Europe, Atmos. Chem. Phys., 18, 2175-2198, doi:10.5194/acp-18-2175-2018, 2018.

Tørseth, K., Aas, W., Breivik, K., Fjæraa, A. M., Fiebig, M., Hjellbrekke, A. G., Lund Myhre, C., Solberg, S., and Yttri, K. E.: Introduction to the European Monitoring and Evaluation Programme (EMEP) and observed atmospheric composition change during 1972–2009, Atmos. Chem. Phys., 12, 5447-5481, doi:10.5194/acp-12-5447-2012, 2012.

Xing, J., Mathur, R., Pleim, J., Hogrefe, C., Gan, C.-M., Wong, D. C., Wei, C., and Wang, J.: Air pollution and climate response to aerosol direct radiative effects: A modeling study of decadal trends across the northern hemisphere, J. Geophys. Res, 120, 12,221-212,236, doi:10.1002/2015JD023933, 2015.

---

## Author Response (AR2)

**Responses to the report of anonymous referee #1**

We would like to thank you for your further comments which helped to improve our manuscript. Please find below your comments in blue, our responses in black and modifications in the revised manuscript related to technical or specific comments in italic and inside quotes. All modifications are highlighted in the revised manuscript.

General Remarks:

Reviewer's response: I am not satisfied with the authors' reply to my first main question. I understand how the authors designed PHOT1-3 when I read their original manuscript. What I do not understand is why the authors did not calculate AOD in 1990 based on the real PM in that time. The purpose of the study is to find the reason for summer surface O3 change between 1990 and 2010 over Europe. The authors used 2010 emission to calculate 2010 AOD. So why would they not do this same in 1990? Of course, the uncertainty of the currently available emissions in 1990 is large as pointed out by the authors. However, the emissions still represent the best estimates by our scientific community. If the authors believe it is more appropriated to give a range of AOD in 1990 in this study, they should at least explain why in the paper.

In your first review you asked "why is it better to use the designed emissions in 1990 instead of using the emissions in 1990 provided directly from emission inventories". Since our intention in this study was not to design 1990 emissions, we tried to explain the reasons in our first reply. We apologize if we misunderstood your comment. We hope to clarify it better in the following.

The purpose of this study is to examine the impact of the European ARI changes between 1990 and 2010 on summer surface ozone. Therefore, we needed to isolate the ARI impact from other factors affecting ozone formation, such as precursor emissions, changes in meteorological and boundary conditions. That is why we did not simulate the actual year of 1990, but using the same input as for the base case (2010), we designed sensitivity tests (PHOT1-3) by scaling up the PM concentrations (**not emissions**) only in the AOD/radiative transfer calculations. In this way we could isolate the impact of changes in ARI on ozone from other factors. If we used a 1990 emission inventory and compared it with the base case (2010), then we would not have been able to disentangle the impact of changes in ARI on ozone from other factors, as the 1990 emissions would have influenced also the chemistry apart from the AOD/radiative transfer calculations. The method that we followed (i.e. with PHOT1-3 scenarios) ensured the isolation of the impact of ARI on ozone and its appropriateness within a realistic range of AOD conditions in 1990, was verified by comparisons of our simulated changes in AOD and SSR with the respective changes in observations (see Sections 4.3.1 and 4.3.2). We hope that the revised version now describes more clearly the purpose of this study along with the details, appropriateness and limitations of the methodology that we

used to address the study's purpose in the last paragraph of the introduction as well as in Section 2.3.1. The following sentences were modified:

From page 4, line 32 to page 5, line 1: *"Third, we modeled and compared only the initial (1990) and final year (2010) of the studied period using same model input (i.e., the one of 2010; thus the actual year 1990 was not simulated to avoid the effects from emissions and meteorology, but rather the AOD and SSR conditions representative of the 1990's were used; see Sect. 2.3) to isolate the influence of ARI on ozone from other factors."*

Page 8, lines 9-12: *"In order to quantify only the changes in ARI, we had to isolate them from other effects such as the gas-aerosol chemical interactions. For this reason, we modified the radiative transfer algorithm in CAMx (i.e. the in-line version of TUV) by applying an adjustment factor ($p_f$) in the AOD calculation to represent the aerosol concentrations in 1990, but without changing the concentrations themselves and thus avoiding any change due to chemistry."*

Page 8, lines 15-17: *"Hence, the product $p_f \cdot C$ represents the PM concentrations in 1990, but purely in AOD calculations in order to generate only AOD, solar radiation and photolysis rates as in 1990."*

Reviewer's response: It is good that the authors have changed the title to address my second main question, although they do not mention it in their reply letter.

We changed the title of the paper following your suggestion but we forgot to mention it in the reply letter. We apologize for this.

Specific question 8

Reviewer's original question: Page 10 lines 4-6: the criteria III should not be listed here since all PM2.5 data in this study are estimated based on PM10, while not measured.

The authors' reply: The criteria iii is about the PM10 and PM2.5 data availability for 2010. In Sect. 2.2 we mentioned that both the PM10 and PM2.5 were provided by Airbase and NABEL networks and both products were measured at the respective sites.

Reviewer's response: It is better for the authors to point out that PM2.5 measurements may not be available in years other than 2010. Otherwise, these sentences are confusing: "they had both PM10 and PM2.5 data for 2010" (page 10 line 20) and "… the aerosol coarse mode was subtracted to estimate PM2.5 concentrations" (page 10 line 22). Readers may not be familiar with the temporal coverage of Airbase and NABEL networks.

We did the following modifications in the manuscript so that the information about the $PM_{10}$ and $PM_{2.5}$ data availability is clearer as well as the reason for calculating the 2010 aerosol coarse mode to infer the trend in $PM_{2.5}$ concentrations:

Page 10, lines 15-16: *"... infer the $PM_{2.5}$ concentrations trend, as there are no $PM_{2.5}$ measurements available for the whole examined period (i.e. from 1990−1992 to 2010), ..."*

Page 10, lines 20−22: *"... i) they covered the whole period 1990–2010 (Switzerland), or 1992–2010 (the Netherlands) for $PM_{10}$ data, ii) they had at least 70% of daily $PM_{10}$ and $PM_{2.5}$ data in each month, and iii) they had both $PM_{10}$ and $PM_{2.5}$ data for 2010 in order to calculate the 2010 aerosol coarse mode ($PM_{10} − PM_{2.5}$)."*

Page 10, line 24: *"... the 2010 aerosol coarse mode was subtracted to estimate $PM_{2.5}$ concentrations as discussed above."*

**Responses to the report of anonymous referee #2**

We would like to thank you for your further comments which helped to improve our manuscript. Please find below your comments in blue, our responses in black and modifications in the revised manuscript related to technical or specific comments in italic and inside quotes. All modifications are highlighted in the revised manuscript.

The authors have taken into consideration most of my comments on the original submission. They now better explain the model set-up and the methodology employed to perturb the photolysis rates. I am also happy that they have clarified that the model simulations do not necessarily represent trends during the 1990-2010 period. Overall, the manuscript is much improved. I have a few additional minor suggestions for the authors to consider:

1) Page 5, line 17: Please clarify whether CAMx employs a modal or sectional approach to represent aerosol size distribution. I believe what the authors imply by "bimodal size distribution" is that 2 sections, nominally representing the fine and coarse aerosol, were used in the model calculations.

In our simulations with CAMx we used a static two-mode (fine/coarse) scheme. We slightly modified the sentence on page 5, lines 17-18 so that is clearer to the reader:

*"… and we simulated the PM concentrations using a static two-mode (fine/coarse) scheme for the aerosol size distribution."*

2) On page 5, line 25 it is stated that TUV employed a climatological aerosol profile. Does that imply that some representation of ARI is already incorporated in TUV? If so, how is this accounted for in the subsequent CAMx photolysis perturbations? Some clarification may be beneficial.

This is correct. First of all we would like to make it clear that the full-science TUV is used as preprocessor to provide CAMx with clear-sky photolysis rates. Then these rates are internally adjusted in CAMx for clouds and aerosols using a fast in-line version of TUV. Yes, there is a representation of ARI already incorporated in TUV. However, the TUV calculations (using a climatological aerosol profile) are repeated inside CAMx (using the in-line version of TUV) as its first step of radiative transfer calculations. In the second step of CAMx radiative transfer calculations, only the aerosol profile that is simulated by CAMx and clouds are taken into account. In other words, the climatological aerosol profile that is used in full-science TUV is also used in the first step of CAMx's radiative transfer calculations, but not in the second step. In general, the purpose of the 2-step calculations in CAMx is to calculate solar fluxes through the grid column for 2 cases: clear sky with default haze (i.e. climatological aerosol profile), and cloudy sky with actual haze (simulated PM). In this way, the ratio of the two solar flux profiles is applied consistently to the clear-sky photolysis rates that are pre-calculated by TUV (Ramboll Environ,

2016). Therefore, the climatological aerosol profile is used consistently between the TUV and CAMx calculations and does not influence the CAMx photolysis perturbations sensitivity tests (PHOT1-3). We did the following modifications in the text for a better clarification:

From page 5, line 24 to page 6, line 1: *"The full-science Tropospheric Ultraviolet and Visible (TUV) radiation model (NCAR, 2011) is used as a preprocessor to provide CAMx with clear-sky photolysis rates, where a climatological aerosol profile determined by Elterman (1968) is used. Then, these rates are internally adjusted in CAMx every hour for clouds and aerosols as well as for pressure and temperature using a fast in-line version of TUV (Emery et al., 2010; Ramboll Environ, 2016). The internal adjustment for clouds and aerosols inside CAMx is performed in two steps: First, the clear-sky radiative transfer calculations with in-line TUV are repeated inside CAMx. In the second step, the radiative transfer calculations are repeated including the impact of clouds and aerosols (simulated by CAMx). A ratio of cloudy- (and aerosols) to clear-sky solar radiation is derived by the aforementioned 2-step radiative transfer calculations in CAMx. This ratio is then applied to the clear-sky photolysis rates and SSR which were calculated by the full-science TUV preprocessor at the beginning. This internal adjustment (i.e., in-line TUV) is carried out only for a single representative wavelength (350 nm), …"*

3) Page 6, lines 15-17: Were the extracted SSR values from TUV and CAMx compared? How does one determine they are consistent?

In TUV we modified the code to extract the **clear-sky** SSR values. In CAMx, the radiative transfer calculations were performed by a fast in-line version of TUV in which: i) the radiative transfer calculations are performed for a single representative wavelength of 350 nm (as also stated in the manuscript), ii) the absorption from $O_2$, $O_3$, $NO_2$, and $SO_2$ was removed since it occurs in narrow UV bands compared to the broad-band influence of clouds, iii) the extraterrestrial flux cancels out in the calculation of cloudy- to clear-sky ratio, and thus was not needed (Ramboll Environ, 2016). In this way, the CAMx runtime is reduced while maintaining the accuracy of the radiative transfer calculations (Ramboll Environ, 2016). Therefore, the ratios of cloudy- (and simulated PM) to clear-sky SSR values from CAMx were applied as a multiplicative factor to the TUV clear-sky SSR values to obtain the final SSR values (also stated on page 5, lines 30-32), which were compared to the BSRN observations. In other words, the comparison between the extracted SSR values from TUV and CAMx is a comparison between clear-sky and all-sky SSR values, respectively. In addition, since the radiative transfer calculations in CAMx are essentially performed by the same TUV algorithm and the aforementioned 2-step approach has been tested and validated for a range of cloudy conditions (Emery et al., 2010), we do not believe that there are inconsistencies in the radiative transfer calculations between TUV and CAMx. We modified Sect. 2.1 (as shown above in our reply to your previous comment) and we included a clarification in the following sentence, so that is clearer that both TUV and CAMx share essentially the same radiative transfer algorithm:

Page 6, lines 15-17: *"In addition, the radiative transfer algorithms of both full-science TUV and CAMx (i.e., in-line TUV) were modified to extract the modeled AOD and SSR data."*

4) In my earlier review I had asked if the authors had examined impacts on daily maximum ozone. The authors responded that the impacts were higher by only ~0.1 ppb and explain that the maximum ARI impacts occur at times different (morning and evening transitions, as shown in earlier studies) from when the daily maximum occurs (mid-afternoon). For completeness, I think it would be useful to point this out in the manuscript discussions

We added a small discussion in the manuscript to address this, from page 14, line 31 to page 15, line 2:

*"We also investigated the impact of ARI changes on daily maximum ozone, but it was higher only by up to ~0.1 ppb (not shown) compared to the daytime (10:00−18:00 LMT) average. Therefore, the ARI did not have a significantly higher impact on daily maximum ozone. The reason is that the daily maximum ozone occurs at different time (mid-afternoon) than the times the maximum ARI occurs (morning and evening), as also shown in other studies (Xing et al., 2015a, 2017)."*

**References**

Emery, C., Jung, J., Johnson, J., Yarwood, G., Madronich, S., and Grell, G.: Improving the Characterization of Clouds and their Impact on Photolysis Rates within the CAMx Photochemical Grid Model. Prepared for the Texas Commission on Environmental Quality, Austin, TX. Prepared by ENVIRON International Corporation, Novato, CA and the National Center for Atmospheric Research, Boulder, CO (August 27, 2010). 2010.

Ramboll Environ: User's guide to the Comprehensive Air Quality Model with Extensions (CAMx). Version 6.3, http://www.camx.com, 2016.